# Evolutionary history and past climate change shape the distribution of genetic diversity in terrestrial mammals

Spyros Theodoridis [1✉], Damien A. Fordham [1,2], Stuart C. Brown [2], Sen Li[1,3], Carsten Rahbek[1] & David Nogues-Bravo[1✉]

Knowledge of global patterns of biodiversity, ranging from intraspecific genetic diversity (GD) to taxonomic and phylogenetic diversity, is essential for identifying and conserving the processes that shape the distribution of life. Yet, global patterns of GD and its drivers remain elusive. Here we assess existing biodiversity theories to explain and predict the global distribution of GD in terrestrial mammal assemblages. We find a strong positive covariation between GD and interspecific diversity, with evolutionary time, reflected in phylogenetic diversity, being the best predictor of GD. Moreover, we reveal the negative effect of past rapid climate change and the positive effect of inter-annual precipitation variability in shaping GD. Our models, explaining almost half of the variation in GD globally, uncover the importance of deep evolutionary history and past climate stability in accumulating and maintaining intraspecific diversity, and constitute a crucial step towards reducing the Wallacean shortfall for an important dimension of biodiversity.

[1] Center for Macroecology, Evolution, and Climate, GLOBE Institute, University of Copenhagen, 2100 Copenhagen Ø, Denmark. [2] The Environment Institute and School of Biological Sciences, University of Adelaide, 5005 Adelaide, Australia. [3] Evolutionary Genomics Section, GLOBE Institute, University of Copenhagen, 1350 Copenhagen K, Denmark. ✉email: spyros.theodoridis@sund.ku.dk; dnogues@sund.ku.dk

The description of the global pattern of multiple dimensions of biodiversity, from intraspecific genetic variation to taxonomic and phylogenetic diversity (PD), is vital both for assessing the underlying processes shaping the distribution of life on Earth and for maximizing the overall protection of biodiversity[1,2]. Increased data availability in past decades on species distributions and their evolutionary relationships, particularly for mammals, has permitted global-scale evaluations of the distribution of two primary dimensions of biodiversity, namely species richness (SR) and PD[3], allowing for their joint consideration as conservation targets[4,5]. These advances have also enabled in-depth assessments of how large-scale eco-evolutionary processes (speciation, extinction, and dispersal) shape latitudinal diversity gradients[6–8] and the mismatches between the spatio-temporal patterns of SR and PD[9,10].

However, because georeferenced genetic data are highly fragmented, spatially and taxonomically[11–14], knowledge of the global distribution of intraspecific genetic diversity (GD), at fine scale, and its covariation with other biodiversity dimensions remains scarce and elusive, contributing to significant gaps of knowledge on the distribution of life, also known as the Wallacean shortfall. A recent global assessment of the covariation between GD and species diversity in fish has shown a weak, yet significant, relationship[14], while regional studies have provided contrasting results, from no covariation in plants[15] and freshwater fish[16] to considerable overlap in coral reef fish[17] and amphibian and reptile assemblages[18]. This has, thus, precluded any generalization regarding the link between intra- and interspecific diversity at global scale and across phylogenetic clades. Given that genetic variation is the raw material of species adaptive potential[19], and ecosystem resilience[2], there is a pressing need to establish whether a general relationship between GD and interspecific diversity exists, and predict its global distribution, particularly in regions of the world that lack sufficient information. Overcoming the Wallacean shortfall for GD provides unique opportunities both in revealing the mechanisms that shape biodiversity in space and time, and in enhancing our predictions of how GD might change under ongoing and future environmental change[20].

Multiple hypotheses, seeking to explain the global biodiversity gradients, have suggested a positive relationship between intra- and interspecific biodiversity, and particularly the spatial congruence between GD and SR/PD[8,21–24]. Early to mid-twentieth century evolutionary biologists suggested that the observed differences in the spatial distribution of taxonomic diversity can be explained by differences in the rates and interaction of evolutionary processes at the intraspecific level, such as mutation, natural selection and genetic drift, through deep evolutionary time[25,26]. Accordingly, the evolutionary speed hypothesis theorizes that increased mutation rates driven by higher temperatures and solar radiation in the tropics, shorter generation times, and the resulting accelerated selection rates lead to faster rates of within-species genetic divergence at lower latitudes[21,27]. The accelerated intraspecific divergence further increases the accumulation rates of species, resulting both in higher GD and SR compared to temperate and polar regions[24,27]. This theory was primarily formulated for ectothermic organisms[21,25], whereas it was hypothesized that evolutionary rates, particularly mutation rates, in endotherms, such as mammals, will not vary systematically with latitude[22]. More recent integrative interpretations of the evolutionary speed hypothesis incorporate the role of increased productivity of tropical biomes in accelerating genetic evolution, expanding this theory to endotherms[27,28]. Additionally, the faster evolutionary rates in tropical biomes, combined with the older age/longer persistence of these biomes and their clades (that is, the time and area hypothesis), are hypothesized to

have allowed higher accumulation of mutations along these clades[22,24], resulting in higher PD and GD in the tropics compared to temperate and boreal biomes. In agreement with the faster tempo of evolution at lower latitudes, the Red Queen theory suggests that the increased metabolic and evolutionary rates in the tropics accelerate rates of ecological interactions and coevolution among species at different trophic levels[23]. These processes favour the maintenance of high GD through selection[29], which in turn generate and maintain high taxonomic diversity at lower latitudes[23,27]. Under the above primary, and largely complementary, theories, the predicted accumulation of speciation events and biotic interactions are hypothesized to further increase evolutionary rates below the species level, introducing a feedback mechanism that maintains high diversity both at inter- and intraspecific levels (diversity begets diversity) through time[23,27,30]. However, explicit tests of these theories on the global spatial covariation between GD and interspecific diversity in endotherms are missing.

Current-day patterns of GD are also postulated to be the result of Late Quaternary climate change, and in particular of the last postglacial temperature fluctuations, through its effects on species demographic processes[31,32]. During the Late Quaternary, frequent variations in precipitation (and the resultant fluctuations in suitable habitat) at lower latitudes are proposed to have driven population isolation and adaptive divergence[33,34]. On the contrary, relative long-term climate stability, both in temperature and precipitation, during periods of large magnitude and rapid climate change, is hypothesized to have provided a climatically stable environment in the tropics, thus enabling population persistence[31,35]. The joint effect of these two climate-driven processes at lower latitudes, that is, higher population divergence due to frequent precipitation variability and population persistence due to long-term climate stability, could result in higher accumulation of GD compared to higher latitudes. In higher latitudes, rapid climate change at decadal to centennial time scales during the last deglaciation may have driven population contractions and extirpations followed by expansions and recolonization of ice-free regions, potentially reducing GD[31,32]. At more recent time scales, human activities (particularly in the past 500 years) have caused wide-scale population declines and extirpations across taxonomic groups[36,37], potentially reducing intraspecific diversity[38]. Yet, a dearth of empirical evidence on the long-term impacts of human-driven land-use change on GD remains[13].

Here, we test the relative importance of the primary hypothesized drivers of the global spatial pattern of GD. We focus on terrestrial mammals because this is the most data-rich animal class regarding spatial and taxonomic coverage of genetic data at a global extent[11,13]. Furthermore, the taxonomic diversity of mammals is well defined[39]. We enriched previously-published global-distribution genetic data[11], and georeferenced and utilized a total of 46,965 mitochondrial sequences for >1500 species (24,395 sequences for cytochrome b [cytb], ~85% increase; 22,570 for cytochrome oxidase 1 [co1], ~15% increase), the largest GD dataset assembled so far for terrestrial mammals (much larger then recently published studies[11,13]). Due to their advantageous properties (e.g. rapid evolution and low recombination) in mammals, these two genetic markers have been extensively used in taxonomic, phylogenetic and phylogeographic studies[40], and therefore constitute the richest resource of genetic data with available spatial information. We used this extensive dataset to map GD globally at equal-area grid cells (385.9 km × 385.9 km). To account for phylogenetic relationships among species in our models, we further mapped GD at Wallace's zoogeographic regions for mammals[41]. Each region was constructed to maximize phylogenetic relatedness among taxa, resulting in areas of distinct

evolutionary history[41]. We analysed the global covariation between GD and SR/PD, while considering for the potential spatiotemporal effects of climate and human-driven environmental change. Climate stability was estimated using two alternative measures that capture complementary information on climate change, the rate of change (centennial trend indicative of long-term change) and inter-annual variability (standard deviation around the trend indicative of centennial variation) in temperature and precipitation, during periods of rapid change in global mean temperature since the Last Glacial Maximum[42–44]. Human-driven environmental change was modelled as the timing of significant historical land-use between 8 kya and industrialization[45], and as measured current-day land-use[46]. We found a strong positive correlation between GD, and PD and SR across the two spatial scales and genes. We also identified significant positive effects of past precipitation variability, and negative effects of long-term temperature and precipitation change and temperature variability in the spatial pattern of GD. The best predictors were then used to map the spatial distribution of GD in mammalian assemblages for both genetic markers globally.

## Results

**GD correlates with interspecific diversity**. Our results show that the distribution of GD in terrestrial mammals positively covaries with other dimensions of biodiversity at the interspecific level (Figs. 1 and 2). At the finest spatial scale (that is, data-rich grid cells; *cytb*: $n = 185$; *co1*: $n = 76$; see below), PD shows the highest independent contribution (IC; the percentage of the explained variance in GD accounted for by each independent variable after

excluding non-significant variables; see Methods) (Fig. 1, Supplementary Table 5) in explaining the spatial distribution of GD across both genetic markers (*cytb*: IC = 49.5%; *co1*: IC = 37.9%) followed by SR (*cytb*: IC = 40%; *co1*: IC = 26%). Importantly, linear models (and the modified *t*-test of spatial association) and multimodel inference (sum of Akaike weights [SAW]) confirm the high positive correlation between GD and PD, and the prominent role of PD in explaining global GD patterns across data-rich cells and genes (*cytb*: SAW = 1, $R^2 = 0.39$, adjusted $P < 0.001$; *co1*: SAW = 0.94, $R^2 = 0.34$, adjusted $P = 0.0325$) (Fig. 1, Supplementary Fig. 4, Tables 1 and 2, Supplementary Tables 3, 4 and 6). The correlation between GD and SR in linear regressions is weaker for both markers, and marginally significant for *co1* (*cytb*: $R^2 = 0.35$, adjusted $P < 0.001$; *co1*: $R^2 = 0.27$, adjusted $P = 0.0546$) (Fig. 2, Supplementary Fig. 4, Supplementary Tables 3–6). The low SAW for SR using multimodel inference (*cytb*: SAW = 0; *co1*: SAW = 0.057) reflects its high correlation with PD (Supplementary Table 1) and its exclusion from the most parsimonious models as a redundant variable (Tables 1 and 2).

When considering the spatial aggregation of related phylogenetic clades within zoogeographic regions (Supplementary Fig. 3), only PD shows a significant contribution and is positively associated with GD for *cytb* ($n = 34$; IC = 34.3, SAW = 0.89, $R^2 = 0.2$, $P = 0.008$) (Fig. 1, Table 1, Supplementary Fig. 5a). None of the considered biodiversity dimensions is identified as significant for *co1* (Fig. 1). The low correlation between GD and PD and SR for *co1* at this coarse spatial scale reflects the highly unbalanced distribution of available *co1* sequences across regions and taxa (Supplementary Figs. 1–3).

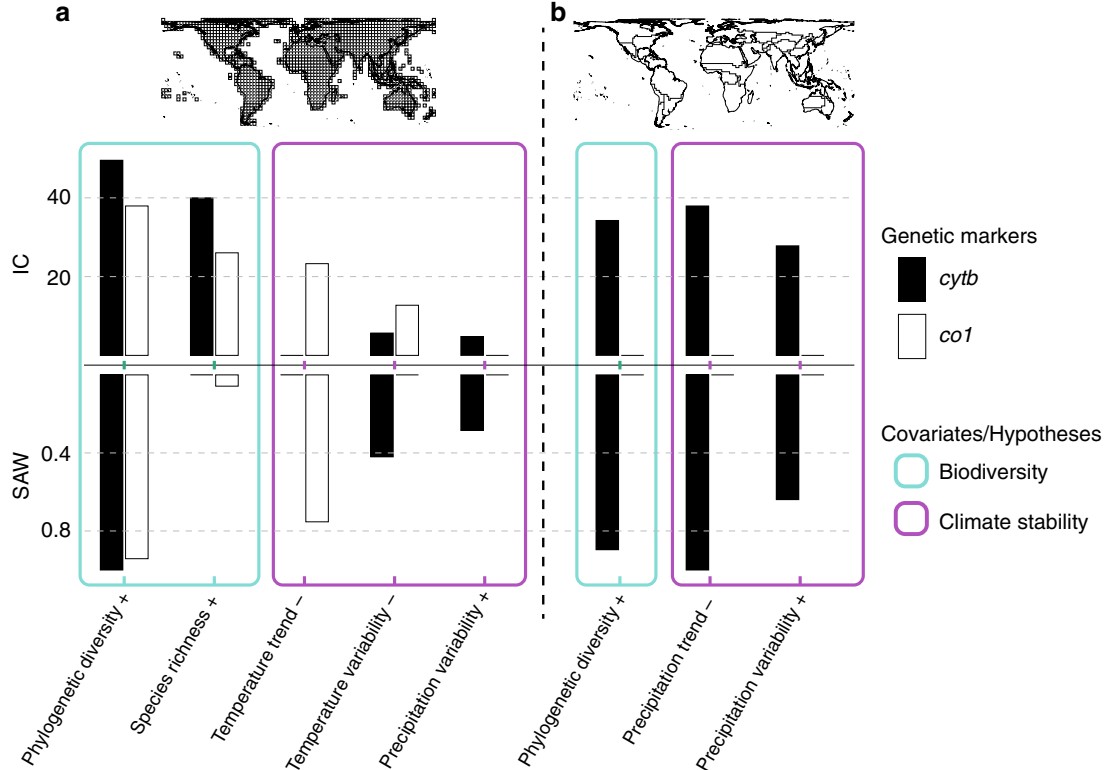

**Fig. 1 Covariates of genetic diversity in terrestrial mammals at two spatial scales.** Independent contribution (IC; hierarchical partitioning) and the sum of Akaike weights (SAW; multimodel inference) of biodiversity dimensions and climate variables in explaining the global distribution of genetic diversity (average number of mutations per base pair and across taxa) for *cytb* and *co1*. Spatial scales: **a** grid cells (*cytb*: $n = 185$; *co1*: $n = 76$), and **b** zoogeographic regions (*cytb*: $n = 34$; *co1*: $n = 30$). Inferences at the grid cell scale are based only on data-rich cells (see Fig. 2 and Methods). Plus (+) and minus (−) signs after each explanatory variable indicate positive and negative significant association respectively between the explanatory variable and genetic diversity. Note that all explanatory variables for *co1* at the zoogeographic scale were insignificant, reflecting the highly unbalanced distribution of available *co1* sequences across regions and taxa (Supplementary Figs. 1–3).

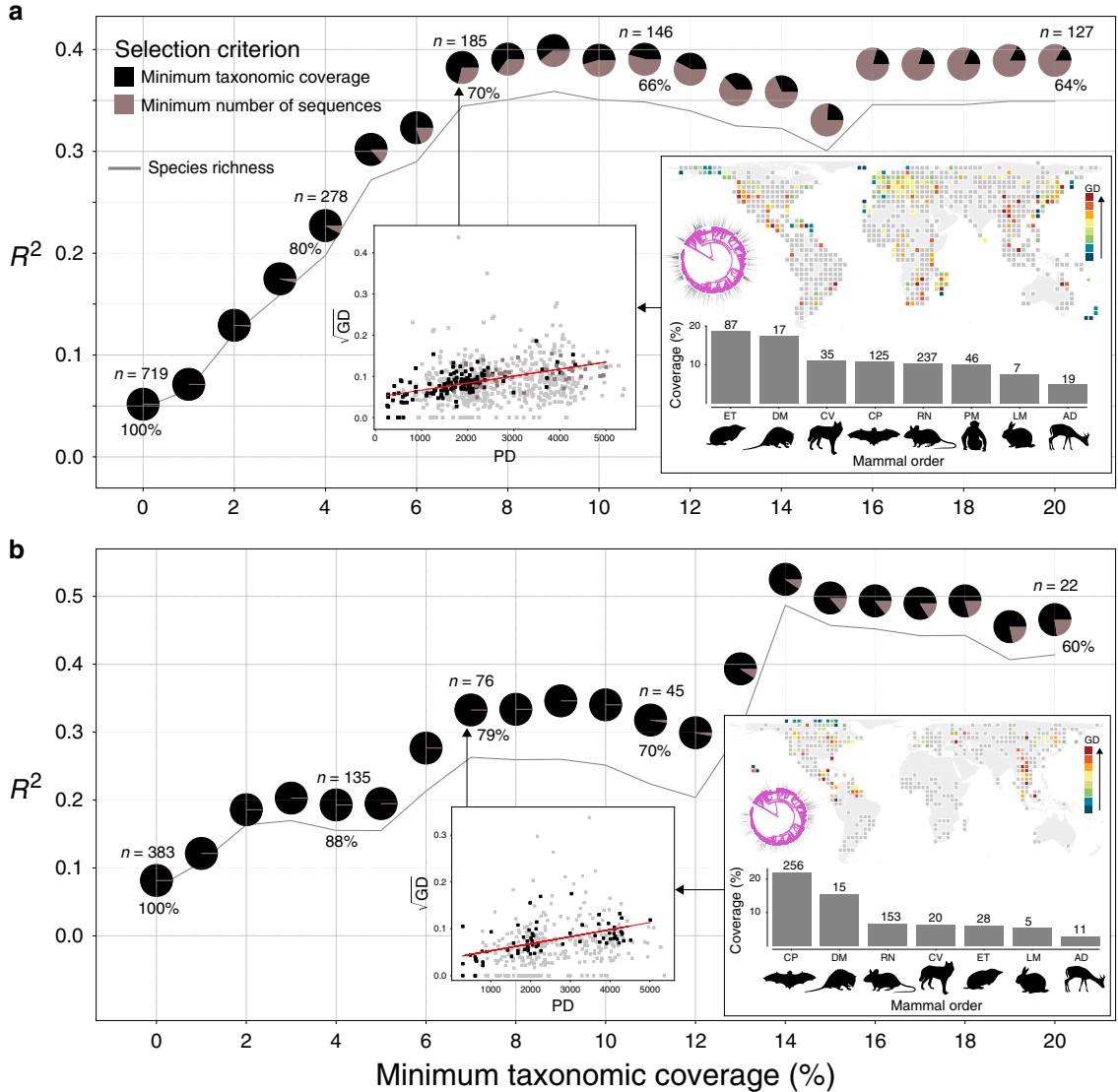

**Fig. 2 Correlation between biodiversity dimensions and genetic diversity at the grid cell scale.** Correlation ($R^2$) between genetic diversity (GD; average number of mutations per base pair) and phylogenetic diversity (PD; millions of years; pies) and species richness (SR; number of species; grey line) for *cytb* (**a**) and *co1* (**b**) across subsets of data when grid cells with low taxonomic coverage and sequence availability are excluded (only cells with a minimum of 55 sequences for *cytb* and 278 sequences for *co1* were retained; see Methods; see also Supplementary Tables 3 and 4 for modified *t*-tests of spatial association). The numbers above and below pies represent the number of retained cells (*n*) and the percentage of retained sequences relative to the full dataset, respectively. Colours in each pie (black and brown) represent the proportion of grid cells that follow each of the selection criteria relative to the total number of retained cells. The inset scatterplot shows the relationship between PD and the square root of GD (grey squares show the full dataset), the map shows the distribution of cells, black lines on the mammalian phylogenetic trees show the retained sequence number (log transformed) per species, and bars show the mammalian order coverage (orders with five or more retained species; retained species numbers on top of the bars). Order abbreviations: AD Artiodactyla, CV Carnivora, CP Chiroptera, DM Didelphimorphia, ET Eulipotyphla, LM Lagomorpha, PM Primates, RN Rodentia.

The correlation between GD and PD and SR increases rapidly for both analysed genes when grid cells with low taxonomic coverage and sequence availability are excluded (Fig. 2, Supplementary Tables 3 and 4), ranging for PD from $R^2 = 0.05$ to 0.4 for *cytb* (0.05 to 0.36 for SR), and $R^2 = 0.08$ to 0.52 for *co1* (0.06 to 0.49 for SR). While data-poor cells represent the majority of cells in the complete data sets for both markers, the retained subsets we use in our analyses (minimum taxonomic threshold = 7% or minimum number of sequences) contain the majority of the entire number of sequences (70% of the total number of sequences for *cytb* and 79% for *co1*) (Fig. 2). This pattern indicates an important difference in efforts to sample GD and geographically annotate these samples across the globe. For example, for the majority of the tropical and subtropical regions

in south America and Central Africa a very low number of georeferenced sequences is available, particularly for *co1* (see Supplementary Figs. 1–3).

**Past climate stability shapes GD**. We defined climate stability both in terms of long-term rates of change (regression slope) and in terms of short-term/inter-annual variability (standard deviation of regression residuals), within centuries of rapid changes in global mean temperature during and after the last deglaciation. We found that, at the grid cell scale, greater inter-annual temperature variability is associated with low GD for *cytb* (IC = 5.7%, SAW = 0.42, $R^2 = 0.07$, $P = 0.002$) (Fig. 1, Table 1, Supplementary Fig. 4a, Supplementary Tables 5 and 6). Additionally, for *co1*,

**Table 1 Results of the multimodel inference across two spatial scales for *cytb*.**

| Scale | Independent variables | AICc | $R^2$ | $R^2$ adj. | $\Delta$AICc | wAICc | Moran's $I$ | P value (Moran's $I$) |
|---|---|---|---|---|---|---|---|---|
| Grid cells | Phylogenetic diversity | −853.15 | 0.385 | 0.382 | 0 | 0.403 | −0.008 | 0.438 |
| | Phylogenetic diversity + temperature variability | −852.63 | 0.39 | 0.384 | 0.516 | 0.311 | −0.006 | 0.425 |
| | Phylogenetic diversity + precipitation variability | −851.5 | 0.387 | 0.38 | 1.652 | 0.176 | −0.01 | 0.458 |
| | Phylogenetic diversity + temperature variability + precipitation variability | −850.52 | 0.39 | 0.38 | 2.628 | 0.108 | −0.013 | 0.48 |
| Zoogeographic regions | Phylogenetic diversity + precipitation trend + precipitation variability | −270.707 | 0.443 | 0.387 | 0 | 0.535 | 0.0561 | 0.24 |
| | Phylogenetic diversity + precipitation trend | −269.916 | 0.381 | 0.341 | 0.789 | 0.361 | 0.06 | 0.228 |
| | Precipitation trend + precipitation variability | −267.428 | 0.334 | 0.291 | 3.278 | 0.103 | 0.125 | 0.126 |

All possible combinations of variables identified as significant from the hierarchical partitioning approach (see Supplementary Table 5) were evaluated using linear models. Models that contained highly collinear variables (Pearson's |r| > 0.7) were excluded. The ranking of the models is based on the corrected Akaike's information criterion (AICc). We retained only models with $\Delta$AICc $\leq$ 5 compared to the best model ($\Delta$AIC = 0) for each spatial scale. Residual spatial autocorrelation was computed for each retained explanatory model across spatial scales using Moran's $I$ and 10,000 simulations (P value).

both large temperature trend (steeper positive slopes indicative of rapid warming) (IC = 23.3%, SAW = 0.75, $R^2$ = 0.29, $P < 0.001$) and high temperature variability (IC = 12.7%, SAW = 0.06, $R^2$ = 0.22, $P < 0.001$) are associated with low values of GD (Fig. 1, Table 2, Supplementary Fig. 4, Supplementary Tables 5 and 6). Conversely, greater inter-annual precipitation variability is positively associated with higher GD for *cytb* (IC = 4.79%, SAW = 0.28, $R^2$ = 0.07, $P < 0.001$) (Fig. 1, Supplementary Fig. 4a, Supplementary Tables 5 and 6). At the scale of zoogeographic regions, only the two components of stability in precipitation (that is, trend and variability) show a significant contribution in shaping GD and can jointly explain about one-third of the variation in GD for *cytb* ($R^2$ = 0.29) (Fig. 1, Table 1). At this scale, and in agreement with the grid cell scale, higher inter-annual precipitation variability is positively associated with higher GD (IC = 27.8%, SAW = 0.64, $R^2$ = 0.19, $P$ = 0.001), while large precipitation trend (that is, steeper positive slopes indicative of rapid increases in precipitation) is strongly associated with low GD (IC = 37.9%, SAW = 1, $R^2$ = 0.2, $P$ = 0.009) (Fig. 1, Table 1, Supplementary Fig. 5a, Supplementary Tables 5 and 6).

**The human footprint in global GD patterns.** Neither the timing of substantial historical land modification, nor the intensity of contemporary human-driven land-use, significantly contributed to our models of GD across the two genes and the two spatial scales (that is, grid cells and zoogeographic regions; Fig. 1, Supplementary Figs. 4 and 5).

**Towards reducing the Wallacean shortfall.** We used the best model (most parsimonious) identified based on multimodel inference (see Methods) for each genetic marker to predict and map the global distribution of GD at the grid cell scale. For *cytb* the best model included only PD and explained 39% of the global variation in GD (Table 1), while for *co1* the best model included PD and temperature trend and explained 38% of the global variation in GD (Table 2). In both cases, we detected no significant spatial autocorrelation in model residuals (*cytb*: Moran's $I = -0.038$, $P = 0.38$; *co1*: Moran's $I = -0.008$, $P = 0.438$; Tables 1 and 2; Supplementary Figs. 6 and 7). Both models predict high levels of GD in grid cells with high PD, particularly in the tropical mountains (Fig. 3), while the high number of grid cells with low GD values for *co1* at northern latitudes is indicative of the effects of higher rates of past temperature change (trends), during periods of rapid global climate change since the last glacial maximum, in these regions (Fig. 3b).

## Discussion

Our results from the two mitochondrial genes sampled from >1000 species show that multiple dimensions of biodiversity, from genetic variation within species, to species and clade diversity, covary globally for mammal assemblages. Furthermore, they provide evidence for a significant role of rapid climate change, both in temperature and precipitation, in reducing GD and centenial precipitation variability in increasing GD globally. When PD and past climate change are considered jointly in statistical models, almost half of the global variation in GD can be explained, allowing for predictions in data-poor regions of the planet. These predictions constitute a first step towards overcoming the Wallacean shortfall for GD, and can inform and be further validated by field-work campaigns in data-poor regions of the Earth.

Our findings of a strong positive correlation between GD and PD/SR at the grid cell resolution (Figs. 1 and 2) are in accordance with predictions from multiple existing hypotheses linking intraspecific genetic evolution at shallow evolutionary time scales

**Table 2 Results of the multimodel inference at the grid cell scale for *co1*.**

| Independent variables | AICc | $R^2$ | $R^2$ adj | ΔAICc | wAICc | Moran's *I* | *P* value (Moran's *I*) |
|---|---|---|---|---|---|---|---|
| Phylogenetic diversity +temperature trend | −322.95 | 0.377 | 0.36 | 0 | 0.695 | −0.038 | 0.38 |
| Phylogenetic diversity | −320.88 | 0.341 | 0.332 | 2.07 | 0.246 | −0.008 | 0.44 |
| Species richness + temperature trend | −317.97 | 0.335 | 0.317 | 4.97 | 0.057 | 0.002 | 0.389 |

All possible combinations of variables identified as significant from the hierarchical partitioning approach (see Supplementary Table 5) were evaluated using linear models. Models that contained highly collinear variables (Pearson's |r| > 0.7) were excluded. The ranking of the models is based on the corrected Akaike's information criterion (AICc). We retained only models with ΔAICc ≤ 5 compared to the best model (ΔAIC = 0). Residual spatial autocorrelation was computed for each retained explanatory model across spatial scales using Moran's *I* and 10,000 simulations (*P* value). Note that all explanatory variables for *co1* at the zoogeographic scale were insignificant.

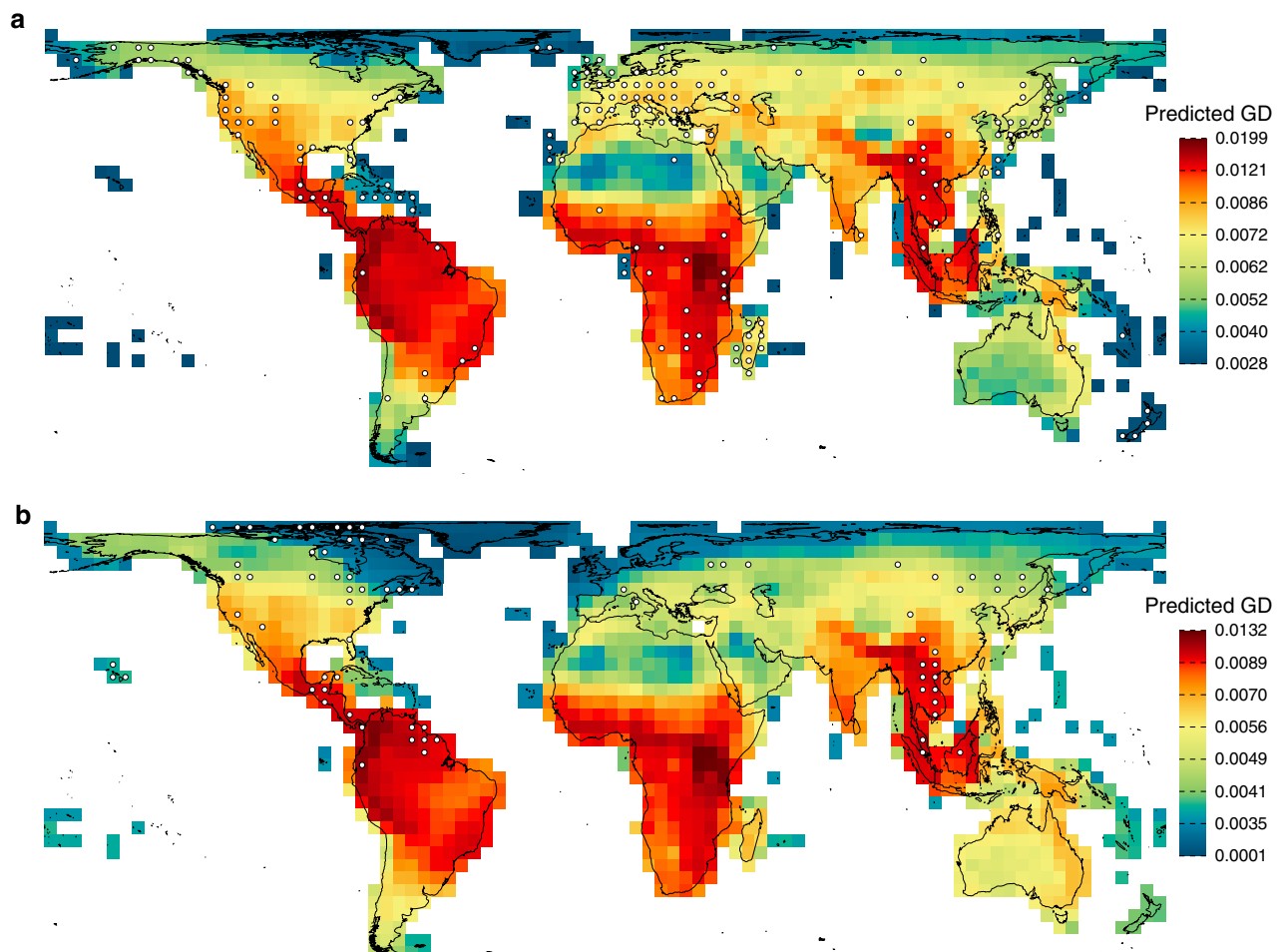

**Fig. 3 Predicted genetic diversity for two mitochondrial genetic markers.** Predicted genetic diversity based on the most parsimonious model identified using multimodel inference (see Tables 1 and 2). For *cytb* (**a**), the most parsimonious model included only PD (see also Fig. 2a), while for *co1* (**b**) the most parsimonious model had phylogenetic diversity and temperature trend as explanatory variables. White dots indicate the retained grid cells (*n* = 185 for *cytb* and *n* = 76 for *co1* at 7% minimum taxonomic coverage) used to build the models.

(microevolution) to evolution above the species level at deep evolutionary time scales (macroevolution)[24]. Here we discuss three basal and frequently invoked hypotheses, that is, the evolutionary speed, the time and area, and the Red Queen hypotheses, that make explicit and complementary predictions regarding the global positive correlation between GD and interspecific diversity. We show that high average number of intraspecific mutations in mammal assemblages coincide with high SR (and PD), providing strong support for the evolutionary speed hypothesis. Previous assessments of the evolutionary speed hypothesis have relied primarily on comparisons between closely related species/lineages, using the rates of molecular evolution

inferred from phylogenies as proxies for faster evolutionary rates at lower latitudes. These studies reported elevated substitution rates towards the tropics both for ectotherms, such as plants[47], marine fish[48], and amphibians[49], and for endotherms, such as mammals[28] and birds[50] (but see ref. [51] and references therein for contrasting results). Additionally, a recent assessment of the continuum between micro- and macroevolution in ectotherms (that is, marine and freshwater fish)[14] reported a weak, yet significant positive relationship between GD and SR globally, partly confirming the expectations of the evolutionary speed theory. In our study, the observed strong association between high GD and high interspecific diversity in tropical regions may further be

explained by the intermediate effect of higher energy-driven productivity of these regions on rates of genetic evolution (potentially through increased carrying capacity) and speciation[27]. Moreover, our results on the dominant role of PD in explaining global patterns of GD are consistent with the time and area hypotheses. The subtropical and tropical biomes, being older, historically larger and climatically more stable through deep evolutionary time, compared to the relatively younger and unstable temperate and boreal biomes, may have permitted older lineages to persist and more diversification to occur within these lineages[22]. Both the age of tropical clades (time) and the area of tropical biomes may have provided more opportunities for mutations to accumulate through time[24], resulting in a positive association between evolutionary time, reflected in PD, and GD. Finally, our results are also consistent with the Red Queen hypothesis that proposes that the highly-diverse tropical ecosystems enhance biotic interactions and selection rates, promoting higher intraspecific GD and population divergence[23,29], and ultimately higher species number, factors that interact and maintain high intra- and interspecific diversity in the tropics[23]. While the spatial resolution of our genetic data does not allow for detailed inferences at the population level, our findings on assemblages of terrestrial mammals strongly suggest that regions of the world that harbour high intraspecific GD coincide with higher taxonomic and PD, providing further support to the above theories on the evolutionary continuum in global biodiversity patterns.

Our results further suggest that Late Quaternary climate change has an important effect on the global pattern of mammalian intraspecific diversity through its effects on species demographic processes, such as population expansion and contraction and the resulting extirpations and divergence. We show that high past inter-annual precipitation variability correlates to higher levels of GD for *cytb*, particularly at the zoogeographical scale (Fig. 1). These results corroborate expectations that spatially and temporally heterogeneous patterns of precipitation in the tropics have caused rapid range fragmentations[33,34], resulting in population isolation and local adaptation, and eventually higher local and regional intraspecific diversity during the Late Quaternary. On the contrary, we show that higher rates (trend) of climate change, that is, rapidly rising temperatures (grid cells; *co1*) and rapid increases in precipitation (zoogeographic regions; *cytb*) during periods of world-wide rapid temperature change since the Last Glacial Maximum, contribute to lower mammal GD in temperate and polar regions. Relatively stable climate trends in tropical regions (e.g. Amazonia and the Andes)[42] are likely to have prevented large demographic fluctuations, such as range expansions and contractions, and thus local population extirpations[31,35], promoting the maintenance of GD. On the other hand, the effects of rapid changes in temperate and cold regions, reflected in temperature and precipitation centennial trends, on species demographic dynamics were more pronounced[26,31]. At these latitudes, rapid and large-scale range contractions and expansions may have resulted in population extirpations and the subsequent loss of GD[31,32]. Overall, our findings agree with recent reports on the contribution of spatiotemporal variation in precipitation and rapid temperature changes on population diversification and extirpation, respectively, through the Late Quaternary[7].

We found no consistent effect of past and recent land-use on the global distribution of GD across the two considered spatial scales. A modelled decline in GD in threatened vertebrates has been reported to have occurred in the last two centuries[38], yet recent results on the effects of modern land-use on the global spatial patterns of GD remain inconclusive[13]. A likely explanation for the absence of a signal, particularly at finer spatial scale (385.9

km × 385.9 km grid cells), is that the effects of habitat disturbance and population declines on GD are likely to be better captured by finer temporal and species-specific monitoring of intraspecific GD at this scale. However, both the amount and spatiotemporal resolution of currently available georeferenced data across the globe does not enable a rigorous evaluation of the influence of human impacts on GD.

There is still a significant lack of data for many regions of the world, especially within the tropics, on the distribution of intraspecific GD (Fig. 2, Supplementary Figs. 1–3). Recent studies have indicated the existence of a latitudinal gradient in GD across animal classes[11,13,14], yet the significant taxonomic and spatial gaps in georeferenced genetic data did not allow for conclusive inferences on the global distribution and drivers of GD. Given a greater availability of georeferenced genetic data from two genetic markers and from >1000 species of terrestrial mammals, we were able to provide fine-scale support for the predictions of multiple existing hypothesis on the positive relationship between GD and interspecific diversity and the significant role of past climate change. Our models, albeit imperfect due to spatial biases in available data and the potential non-inclusion of relevant evolutionary and/or ecological variables, can explain almost half of the global variation in GD across the two genetic markers. More effort in data collection on the ground and annotation, particularly in data-poor regions, such as south America and Central Africa, will further validate our global inferences and predictions, thus helping filling major knowledge gaps for GD. The predicted global distribution of GD at a grid cell spatial scale may further enhance our capacity to mechanistically model the effects of important evolutionary processes (including mutation, drift, gene flow and natural selection) on global biodiversity gradients[8]. While variation in the two mitochondrial genes used in this study is a good indicator of evolutionary history, shaped by population divergence and extirpations, it may not be representative of the patterns of variation across the genome, primarily shaped by gene flow and natural selection. As genome-wide data become available at finer spatial resolutions, our predictive maps may further serve as a baseline for assessing the role of the above evolutionary processes in driving local and regional deviations from the predicted global patterns of GD[52].

The tropics, which are recognized as evolutionary (phylogenetically)-rich regions of the globe for mammals[3,4], are predicted to host high levels of genetic variation within species (Fig. 3). Moreover, the higher explanatory power of PD compared to SR in explaining the distribution of GD suggests that the total amount of deep-time evolution and the accumulation of phylogenetically divergent taxa (as encapsulated by PD)[9] affects the accumulation of mutations and genetic variation globally. These findings highlight an important role evolutionary history in biodiversity conservation[5] and the need to conserve tropical regions as reservoirs of mammalian GD and evolutionary potential. Our spatial model projections indicate that the Northern Andes, the Eastern Arc Mountains, Amazonia, the Brazilian Atlantic forest, the central America jungles, sub-Saharan Africa and southeastern Asia may be regions harbouring the most genetic variability globally (Fig. 3). These regions are also among the most exposed to anthropogenic threats including deforestation and changes in fire regimes[53]. As global change continues to transform Earth's biota, conserving areas of particular importance for primary dimensions of biodiversity, including genetic variation in wild populations, will help meet the targets of the Convention on Biological Diversity (Aichi Strategic Goal C) for halting accelerating rates of biodiversity loss.

GD represents the biosphere's fundamental information bank, defining life's capacity to persist and evolve in response to global environmental change[54]. The need to better understand key

processes that underlie broad-scale patterns of GD has paved the road for the emergence of the new discipline of macrogenetics[55]. The ever-increasing availability of genome-wide data across space and taxa will provide new and important opportunities not only to improve our understanding of the spatial distribution of genetic variability but also to enhance understanding of the adaptive potential of biodiversity globally in the face of environmental change.

## Methods

**Georeferencing and aligning sequence data**. We obtained mitochondrial sequence data for terrestrial mammals from GeneBank and BOLD. GenBank data were downloaded on 16 May 2017 using the Entrez Utilities Unix Command Line. Data from BOLD were downloaded on 25 May 2017 directly from BOLD webpage using the application-platform interface (API) (http://www.boldsystems.org/index.php/resources/api). Our mammalian database consisted of 110,218 sequences for cytochrome b (cytb) and 43,961 sequences for cytochrome oxidase 1 (co1).

We assigned geographic coordinates to the sequences that were not already annotated with latitude and longitude using the API tool provided by GeoNames.org (http://api.geonames.org). To increase the quality of the data in our georeferenced database, we excluded from this procedure any sequences that had only the country name as locality information. From the full set of georeferenced sequences, we extracted all base pairs corresponding exclusively to cytb or co1 by mapping the sequences against the reference mitochondrial genome of Lepus europaeus (accession number: AJ421471). Sequence mapping was performed in Geneious v8.1.7 (ref. [56]) using default settings. We then extracted all base pairs that were successfully mapped against the homologous cytb and co1 loci of the reference genome. For each unsuccessfully mapped sequence we extracted all cytb and co1 base pairs using customized Shell scripts[11]. After the mapping, we further removed all sequences with IUPAC ambiguity codes. Following these filtering steps, species-specific alignments were generated using default settings in MUSCLE[57]. The enriched georeferenced and aligned dataset included a total of 54,786 mitochondrial sequences for 2128 species (1690 species for cytb, 1153 species for co1), representing overall, 36% of the available cytb and co1 sequences at the time of retrieval.

**Filtering out georeferenced sequences**. As we were only interested in the natural range of extant species, we spatially filtered out sequences outside the known native ranges of species. We first checked the spatial concordance between the georeferenced sequences and species' geographic ranges obtained from the most recent version of IUCN geographical ranges (v6.2) for terrestrial mammals[58]. We used the IUCN species range polygons (ESRI-formatted shapefile) and excluded polygons with presence values of 3 ("possibly extant") and 6 ("presence uncertain"), as these represent highly uncertain estimates of distribution. We also excluded range polygons with origin values of 3 ("introduced") and 4 ("vagrant"). We then compiled a comprehensive database relating the IUCN accepted species names with all available species synonyms for terrestrial mammals. We obtained species synonyms from a previous study[59] and the Integrated Taxonomic Information System (https://www.itis.gov/). For cases where we found no match between IUCN and the sequence databases, we further checked the relevant literature and identified synonyms. In some cases, two or more species from GenBank were matched to one single species in IUCN. These cases mostly reflect recent taxonomic revisions and newly described species not yet assessed by IUCN. For the spatial filtering in these cases, species were treated as one single species, following the taxonomic nomenclature of IUCN, and sequences falling outside the ranges of IUCN accepted species were excluded.

For each sequence, we calculated the minimum distance between the boarders of the respective range polygons and the sequence coordinates, and excluded sequences with a minimum distance >385.9451 km (the selected resolution of the grid cells; see below). The final filtered sequence database included 24,395 sequences for cytb and 22,570 sequences for co1 (Supplementary Figs. 1–3).

**Calculating GD**. We defined GD as nucleotide diversity—that is, the average number of nucleotide differences per site in a pairwise sequence comparison[11]. GD within species was then defined as the average number of nucleotide differences per site across all pairwise sequence comparisons for that species. It is mathematically defined as

$$\hat{\Pi} = \frac{1}{\binom{n}{2}} \sum_{i=1}^{n-1} \sum_{j=i+1}^{n} \frac{k_{ij}}{m_{ij}},$$

where $k_{ij}$ is the number of different nucleotides between sequence $i$ and sequence $j$, $\binom{n}{2}$ represents the number of pairwise comparisons made, and $m_{ij}$ is the number of shared base pairs between sequence $i$ and sequence $j$. We considered only pairwise comparisons where sequences overlap in at least 50% of the longer sequence and ignored all positions with gaps or "N" (unknown nucleotide). For a particular assemblage of species, GD was calculated by averaging nucleotide diversity per site across all species present in that assemblage. Thus, we calculate the GD of the

assemblage $t$ ($GD_t$) with genetic sequences from $S$ species as

$$GD_t = \frac{1}{S} \sum_{p=1}^{S} \hat{\Pi}.$$

We calculated GD at two spatial grain resolutions and for both markers independently:

(1) *Equal-area grid cells*. To map the distribution of GD at a finer scale across the planet, we defined an equal-area grid (Behrmann cylindrical equal-area projection) with cell size of 385.9 km × 385.9 km representing 148,953 km² area grids. This grid cell area has been previously shown to be the most appropriate for maximizing the number of sequences from each species that are included in the calculations of $\Pi$ (Eq. 2) and minimizing the difference in the number of sequences between species[11]. We then assigned each georeferenced sequence to a grid cell and calculated GD of each cell as described above.

(2) *Wallace's zoogeographic regions*. To compare GD across independent evolutionary regions of the planet for terrestrial mammalian species and account for phylogenetic relationships across species, we calculated GD across the updated Wallace zoogeographic regions[41] (34 regions, Supplementary Fig. 3). GD for each region was calculated as described above.

**SR and PD**. We evaluated the global spatial correlation between GD at intraspecific levels and the two main biodiversity dimensions at the interspecific level—that is, SR and PD. To this end, we first identified species ranges intersecting each spatial unit (that is, grid cells and zoogeographic regions) using the IUCN species ranges described above[58]. We then calculated SR as the total number of species in a given spatial unit. To be consistent with the estimates of PD (see below), we estimated SR as the total number of species in a given spatial unit that were also present in the mammalian phylogeny. To estimate PD, we obtained a dated mammalian phylogeny from a recent study[59]. While this phylogeny is the most comprehensive dated phylogeny for mammals to date, we acknowledge that branch length information may be inaccurate, particularly for branches that have not been dated previously[59]. However, this reduction in accuracy is likely to be fairly small and restricted only to a portion of the external branches[59], thus expected to have minor impact in our global estimates of PD (see also ref. [4]). We first calculated PD as the sum of tree branches (branch length is measured in millions of years) connecting all species occurring in each spatial unit[60] for all 1000 trees from the posterior distribution of the published phylogeny. We then estimated PD per spatial unit as the mean PD across all 1000 trees.

**Climate stability**. We calculated climate stability for temperature and precipitation during the most recent inter-glacial period by calculating centennial trends and variability around the trend. We did this because these two measures of stability capture complementary information on low frequency/long-term (that is, centennial trend) and high frequency/short-term (standard deviation [SD]) climate variation[42,43]. Based on theory, we expect that regions with long-term stable climate (low trend in temperature and precipitation) and short-term unstable precipitation (high SD), during periods of rapid change in global mean temperature over the past ~ 21,000 years, will coincide with locations that have experienced less population extinction and population adaptive divergence, thus high GD. Paleoclimate data from the TraCE-21ka experiment were extracted using PaleoView[61] at a monthly time-step between 21,000 BP and industrialization (1850 AD; 100 BP). We first identified past centuries of rapid change in global annual mean temperature (that is, extreme centuries) as those having global mean temperature trends (that is, slopes of the generalized least-squares regression) greater than the 90th percentile of the cumulative distribution function built from the trends within all past centuries (1-year time step between windows)[43]. We then calculated local measures of century time-scale trends for each grid cell ($n = 10,368$ cells, 2.5° × 2.5°) for the identified extreme centuries in global mean annual temperature[42]. We also calculated grid cell estimates of variability, where variability was defined as the standard deviation of the residuals about the local trend[43]. We calculated the median trend and median variability across time (that is, across all extreme centuries of change in global temperature) for both temperature and precipitation. We then extracted the values of the climate grid cells intersecting each spatial unit/polygon (grid cells, zoogeographic regions) and calculated the mean value of each climate variable within each spatial unit.

**Human footprint**. We tested the effects of both historical and recent human land modification on the global distribution of GD. To estimate historical land modification, we used the KK10 anthropogenic land cover change dataset[62] that provides grid cell-based estimates of anthropogenic land-cover change (ALCC). The KK10 dataset quantifies anthropogenic impacts on terrestrial carbon storage by combining a global vegetation model with an empirical relationship between population and land-use based on observed data from a number of European countries over preindustrial time[62]. The model output is used to evaluate the impacts of humans on land-use and its subsequent effects on terrestrial carbon storage during the preindustrial Holocene. The data cover the period from 8000 BP to 100 BP, at a resolution of 0.08° × 0.08°. The model assumes that the physical environment over time is stable (that is, climate or geomorphic change was stable over the Holocene). Despite this major assumption, independent validation (for

Northwestern Europe), using information on vegetation communities from pollen cores, has shown that the model performs well when predicting land-use change through the Holocene at a coarse country-level resolution[63]. To evaluate the impact of ALCC during the Holocene prior to industrialization, we identified the time when each land cell is simulated to have surpassed a 20% threshold of human induced land-cover change. The 20% exceedance threshold for significant impacts has been used before for assessing historic human land-use impacts on the biosphere[45]. For cells that were not simulated to be impacted by human land-use prior to industrialization (e.g. Greenland, Antarctica, some parts of northern Canada), a value of 0 was assigned.

For a more recent (post industrialization) estimate of human-driven land-use change, we used a remotely sensed estimate of the human footprint on the terrestrial environment (~1 km² resolution at the equator) corresponding to the year 2009 (ref. [46]). This variable integrates remotely sensed and bottom-up survey information on the following human pressures: (1) the extent of built environments; (2) crop land; (3) pasture land; (4) human population density; (5) night-time lights; (6) railways; (7) roads; and (8) navigable waterways. These pressures are weighted according to estimates of their relative levels of human pressure[46] and summed together to create a standardized human footprint for all non-Antarctic land areas (values range from 0/no pressure, to 50/maximum pressure).

**Linear regressions and effect of data availability**. We first evaluated the distribution of GD at the grid cell and zoogeographic regions scales. As the distribution of GD at the grid cell scale was highly skewed towards zero, we transformed GD to normality using its square root. All subsequent statistical analysis at the grid cell scale were based on the transformed GD. We kept all eight independent variables in their original units. We then explored the association between the global spatial pattern of GD and the eight independent variables—that is, SR, PD, trend and variability in temperature, trend and variability in precipitation, and historical and recent human footprint—using simple linear regressions across genetic markers (*cytb*, *co1*) and spatial scales (grid cells, zoogeographic regions). Initial analysis at the grid cell scale revealed striking differences in the taxonomic coverage and sequence availability across cells (Supplementary Figs. 1 and 2). Taxonomic coverage is defined as the ratio between the number of species used for estimating GD and the total number of species naturally occurring in each grid cell estimated from IUCN range maps. These differences in data availability are expected to impose biases in our spatial estimates of GD due to limited information in the majority of grid cells. To overcome this limitation, we contacted all statistical analyses using multiple subsets of our full dataset (that is, all cells with GD information) based on two filtering criteria—that is, the minimum taxonomic coverage and a minimum number of sequences per grid cell. We fixed the minimum number of sequences per grid cell to one standard deviation of the distribution of the total number of sequences per grid cell in the full dataset (standard deviation for *cytb* = 55; standard deviation for *co1* = 268). By keeping the data-rich grid cells (high taxonomic coverage or a minimum number of sequences) we were able to reveal the high correlation between GD and SR and PD, as well as the effects of climate stability, in explaining spatial patterns of GD at the finest scale and across both genetic markers (Figs. 1 and 2). The results of the linear regressions across the two scales and genes are given in Supplementary Figs. 4 and 5. We further provide all pairwise correlations among the eight explanatory variables in Supplementary Tables 1 and 2.

Since all the diversity variables show some spatial structure (that is, higher values towards the tropics), and all these variables are measured over the same locations, we further tested the significance of the association between GD and PD and SR using the modified *t*-test of spatial association[64] as implemented in the SpatialPack R package[65]. We tested the spatial association across all the subsets of our full dataset described above (see also Fig. 2) and the results are given in Supplementary Tables 3 and 4.

**Hierarchical partitioning**. We used a hierarchical partitioning approach[66] as implemented in the hier.part R package[67] to test for the significance of the independent contribution of the eight variables on the global spatial pattern of GD. This approach allows the estimation of the independent contribution (IC) of each predictor variable in the total explained variance by considering all possible combinations of explanatory variables using linear regressions[66]. The calculation of IC is based on the improvement of model fit (that is, increased $R^2$) by incrementally adding variables starting from a simple single-parameter regression for a given variable and then averaging the model fit differences over all combinations in which that variable occurs. This approach has advantages over alternative techniques as it allows the joint evaluation of multiple independent variables in the presence of multicollinearity[66,68] (e.g. between SR and PD). We estimated the significance of IC by randomizing the values of each explanatory variable 1000 times to obtain 1000 simulated values of IC. We then estimated the significance of each variable's IC using the 95th percentile values of the simulated IC values and the observed IC value of each variable[67]. To remove the effects of non-significant variables in our final estimates, we applied the hierarchical partitioning approach in an iterative fashion, whereby we started by including all eight variables and at each iteration we excluded the variable that showed the lowest (and insignificant)

contribution. Iterations stopped when all retained variables were significant. Results of the hierarchical partition across the two genetic markers and the two spatial scales are given in Fig. 1 and in Supplementary Table 5. Note that all independent variables for *co1* at the zoogeographic scale were insignificant. Also note that for the grid cell scale we only present results obtained using the data subsets containing data-rich cells both for *cytb* and *co1* (see "Linear regressions and effect of data availability" section).

**Multimodel inference and predicted GD**. To identify the models that best explain the global distribution of GD in terrestrial mammals across genes and spatial scales, and further check for consistency with the IC values obtained from hierarchical partitioning, we applied multimodel inference using the corrected Akaike's information criterion[69] (AICc). To this end, we first fitted linear models for all possible combinations of significant variables retained after the hierarchical partition step and excluded models that contained highly collinear variables (Pearson's |r| > 0.7)[70,71]. Applying this threshold removed the effects of redundant models that included variables within the same explanatory hypothesis (e.g. SR and PD, or temperature variability and temperature trend). Collinearity among variables representing different hypotheses was observed only for *co1* at the grid cell scale (see Supplementary Tables 1 and 2 for collinearity among independent variables). Subsequently, for each model we calculated the AICc, ranked the models according to their AICc, and calculated the difference (ΔAICc) between the AICc value of the top-ranking model and the AICc value for each of the other models. We then excluded models with ΔAICc > 5 as models with higher values are commonly considered uninformative[69,72] (varying this threshold to a maximum of 10 had minor effects on the model selection procedure; results not shown). We then recalculated the Akaike weight for each of the retained models. Finally, for each variable across the retained models we summed the Akaike weights for each model in which that variable appears[69]. The SAW is a proxy for the relative importance of variables under consideration. The results of the multimodel inference across spatial scales are shown in Fig. 1, Tables 1 and 2, and Supplementary Table 6. Like the hierarchical partitioning analysis, we only show the results obtained using data with a minimum taxonomic coverage of 7% or a minimum number of sequences per grid cell for both genetic markers.

To further test the adequacy of linear models in explaining global patterns of GD, we visually inspected the normality in residuals using QQ plots (Supplementary Fig. 7) and tested for residual spatial autocorrelation for each retained explanatory model across spatial scales using Moran's *I* and 10,000 simulations implemented in the python package PySal v.2 (https://pysal.org/). Spatial autocorrelation in residuals would be indicative of violation of the assumption of independently and identically distributed residuals and can increase type I error rates (that is, falsely rejecting the null hypothesis of no effect of the independent variables)[73]. It further indicates failure to account for important spatial processes/variables that induce spatial structuring in the dependant variable[73]. Spatial weights were defined using the simple contiguity criterion for the zoogeographic scale, and the *k*-nearest neighbour criterion with *k* = 4 for the grid cell scale. We detected no significant spatial autocorrelation in model residuals across spatial scales and genetic markers (Tables 1 and 2) and these results were robust regardless of the choice of *k* (one to four).

**Reporting summary**. Further information on research design is available in the Nature Research Reporting Summary linked to this article.

## Data availability
The identifiers for all genetic sequences used in this study and the processed data to recreate the figures are available in https://github.com/spyrostheodoridis/Genetic-geography-of-terrestrial-mammals. An interactive exploration of the reported findings is available through the web application https://geneticgeography.com. The Supplementary Note 1 provides instructions on the functionality of the web application. The raw genetic sequences are available in GeneBank (www.ncbi.nlm.nih.gov/genbank) and BOLD (www.boldsystems.org). Data used for the analysis of past climates are available through the PaleoView software (https://github.com/GlobalEcologyLab/PaleoView). Species range maps are available through IUCN (https://www.iucnredlist.org/resources/spatial-data-download). Dated phylogenies are available in Dryad (https://datadryad.org/stash/dataset/doi:10.5061/dryad.bp26v20)[74]. The anthropogenic land cover change during Holocene is available in PANGAEA (https://doi.pangaea.de/10.1594/PANGAEA.871369)[75]. The human footprint maps are available in Dryad (https://datadryad.org/stash/dataset/doi:10.5061/dryad.052q5)[76]. Data for zoogeographic regions are available in https://macroecology.ku.dk/resources/wallace.

## Code availability
All codes used to calculate GD are available in https://github.com/spyrostheodoridis/Genetic-geography-of-terrestrial-mammals.

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

## Acknowledgements

S.T., D.N.-B. and C.R. acknowledge the Danish National Research Foundation for funding for the Center for Macroecology, Evolution and Climate, grant no. DNRF96. Australian Research Council grants supported D.A.F. and S.C.B. (DP180102392, FT140101192).

## Author contributions

S.T., D.N.-B., D.A.F. and C.R. framed the study. S.T., S.L. and S.C.B. carried out the analyses. S.T., D.N.-B., D.A.F. and C.R. discussed and interpreted the results. S.T., D.N.-B. and D.A.F. wrote the manuscript.

## Competing interests

The authors declare no competing interests.
