## [Peer Review File · Nature Communications]

Reviewers' Comments:

Reviewer #1:

Remarks to the Author:

Review : « Spatiotemporal drivers of genetic diversity in terrestrial mammals »

In the manuscript « Spatio temporal drivers of genetic diversity in terrestrial mammals », Theodoridis and colleagues collected >50000 mt sequences for >1500 species to better describe the spatial covariation of the intraspecific genetic diversity, species richness and phylogenetic diversity of mammals at global scale considering the effect of Late Quaternary climate change and land-use change during the Holocene.

I congratulate the authors for the effort of assembling such large datasets. The novelty in their study is to consider phylogenetic diversity, past climatic and land use changed at global scale. This is an important topic to better understand the pattern of genetic diversity at global scale, a level of biodiversity that just start to be considered. I appreciate the effort done by the author. However, I think that in its current form, the paper is mainly descriptive, and requires strong clarifications on the statistical analysis. We are now needed more than a descriptive papers in the field of global spatial and temporal patterns of genetic diversity after the seminal papers of Miraldo et al (2016) and Millette et al. (2019). If the authors are able to relate more directly their results to theory and assumptions, and to clarify, justify or modify their statistical analyses, the paper can have potential to be published. But it requires a major revision and clarifications about the usefulness of the predictions. But if the authors are not able to clarify the utility of the predictions (Fig 3 and 4), I am effraid that the study lack of original results to be published in Nature Communications. I summarize below my main comments. Most of my comments can be found in a marked version of the manuscript.

-My first point is related to the need to clarify the novelty of the paper in regards of the two published papers of Miraldo et al. (2016) and Millette et al. (2019) that alos included mammals mammals? Why their database is largest since the number of sequences for mammals seems to be smaller (see my comment in the text). I assume that it has been updated in the time, and for Millette, only COI is available, but it needs to be clearly explained....This point is related to the need of a clear position of the current work in regards on previous work on the topics, ie recent state of the art on the global patterns on genetic diversity in mammals from at least this two papers.

-A second point is related to the assumptions in the introduction : they are very general. Is it possible to provide more specific assumptions for mammals ? Why having chosen to focus on mammals ? Miraldo et al. included amphibians and Millette et al invertebrates, birds and fishes : they focus on vertebrates. Is there any specific reasons? Probably that the conclusions can be more general if extented to birds and amphibian (ie terrestrial vertebrates) at least. Looking for mammals is fine but it requires some justification and specific assumptions. Related to assumptions, I am expecting a specific assumption for each factor tested. This can help to give a direction to the paper.

-Concerning the main figures and results :

- In figure 1, I feel that it is not necessary to show the ouputs for the both markers since the trends is similar. I suggest to keep the more robust (cytb seems to have a better coverage).
- The Figure 2 is a sensitivity analysis. So it can be moved in supplementary material : it helps to discuss the power of the main analysis but do not bring any new result.
- But more important, is my misunderstanding of the results of the Figure 3 and 4. Probably that I miss something but I do not understand the utility of the predictions. I do not understand why not analyzing directly the relation between genetic diversity and latitude as it has already been done by

Miraldo et al for mammals in 2016 (Fig. 3A) ? Miraldo et al showed that GD was lower at high altitude and higher in the tropics. I feel that the information the authors are looking for in the predictions are already available in the data. But again, it is possible that I do not understand this analysis and its background, and in this case, it requires more explanations.

-I have a quick look on the discussion but as it can strongly change, I did not work more on it. The discussion should not repeat the introduction and should interpret the results.

Finally, I have questions on the statistical method (model):

1) In the linear regression, I am not sure whether the variable GD has been transformed or not before applying linear regression. It is probably not a normal variable. Did the authors check for its distribution ? The linear regression on the raw variables is not appropriated. I potentially understand that a square root transformation has been applied in the grid cell ? Is it true ? Why not at the other spatial scales? Explanations on this point are essential since this is the starting point of the analysis. It needs clarifications in the paper.

2) The authors tested all models and they didn't first check for variables/factors colinearity but they removed models with correlated variables. I am wondering how many models were removed ? I am wondering why not first checking for colinearity and then applying models on uncorrelated variables : it seems a more classical strategy. I assume that variables are highly correlated. Which one? I really need more information on this point, and how correlations do not confound the results.

Yet, all the explanatory variables are measured in different units : if analyzed together, they should probably be normalized?

3) The test of spatial autocorrelation indicated no spatial correlation in their analysis. Can the authors explain more what does this spatial autocorrelation reflect and how it is different from the one discussed in the paper of Gratton et al (2017). Why it is not useful to add a term of spatial autocorrelation in the model ?

Finally, I recommend to avoid to use too many abbreviations. The paper becomes difficult to read.

I was not able to evaluate the sections related to the description of climate stability and Human footprint since I am not a specialist of this field, as well as of phylogenetic diversity. I observed that the two sections in the method describing this part were really more detailed than the other one. I am wondering if those sections were not too detailed or the other methods sections not enough : probably need to find a compromise between both.

I included more comments in a marked version of the paper.

Suggestion of additional biblio related to the topic

Tucker C, Cadotte M. Unifying measures of biodiversity: Understanding when richness and phylogenetic diversity should be congruent. *Diversity and Distributions* 19, 845-854 (2013)

Voskamp A, Baker DJ, Stephens PA, Valdes PJ, Willis SG. Global patterns in the divergence between phylogenetic diversity and species richness in terrestrial birds. *Journal of Biogeography* 44, 709-721 (2017).

Reviewer #2:

Remarks to the Author:

In the manuscript #NCOMMS-19-37347, the authors evaluated >50,000 mitochondrial genetic sequences for >1,500 mammal species to assess the biodiversity, climatic and anthropogenic correlates of genetic diversity worldwide. Moreover, the authors also challenge the Wallacean shortfall for the most basal level of biological diversity. Based on wide databases and sophisticated methods, this study presents an innovative piece of work in an elegant theoretical context. I have just a few comments and suggestions (please, see attached document), which I hope to help the authors improve just details.

Sincerely,

Matheus S. Lima-Ribeiro

Reviewer #3:

Remarks to the Author:

Review of "Spatiotemporal drivers of genetic diversity in terrestrial mammals" by Theodoridis et al.

This manuscript addressing the relationship between genetic diversity based on two genes, phylogenetic diversity, and species richness in mammals globally while also taking into account variation in historical climate since the LGM and both past and present human impacts. The question is interesting and timely, the dataset and analyses are laudable, and the writing is well done. My concerns are as follows:

- 1) The manuscript lacks discussion of or justification for why cytb and co1 were selected, what their significance is, and the extent to which they are representative of genetic diversity more broadly
- 2) The manuscript lack justification for why analyses were conducted using 4 degree latitudinal bands. Biologically, this categorization seems odd given the longitudinal variation within and among continents and the level of environmental heterogeneity with longitude. The most biologically relevant of the three spatial scales of analysis is the zoogeographic regions, but these show the "weakest" results, or at least results that differ from the grid cell and 4 degree latitude bands. Discussion of the results minimizes the geographic regions. I would suggest that the 4 degree band analysis be excluded and the comparison between the grid cell and zoogeographic regions be given more weight. In this context, the relationship between genetic diversity and species richness is much weaker.
- 3) The manuscript is framed largely in a conservation biology context with a secondary focus on the basic evolutionary question at hand. No alternative hypotheses for a positive relationship between genetic diversity and phylo/richness are presented or explored, which essentially sets to paper up as confirmatory science rather than as a test of the relationship between genetic diversity and phylo/richness. More discussion of conflicting results in the literature and the ways in which this analyses does (or does not) resolve them is needed, particularly given the fact that mammals may differ from other taxa in which differing results have previously been found.

Minor comments:

Line 12: Clarify that the covariation is positive.

Line 15: Replace "the" information gap with "an" information gap and clarify that the most basal level of biological diversity is genetic (rather than taxonomic, for example).

Line 27-28: Tell us more about the opposing evidence and what might explain differences

Line 43: long-standing theories, such as??

Line 51-54: This would be a place to add alternatives

Line 71-72: Why these two?

Line 114: Why the square root?

Line 125-126: Consider adding the spatial distribution of these as a supplement to assess how overall results may be affected by geographic sampling biases

Line 129: It's unclear what the biological significance of the latitudinal bands is.

Line 137: What happens if you limit the geographic extent to only the well represented ones? Biologically this is the most relevant measure.

Line 141: Rephrase section head.

Figure 1: Add sentence to main text defining/justifying focus on IC and SAW and which is a better measure (b/c results differ, particularly for importance of species richness)

Figure 2: Axes labels for genetic diversity and phylogenetic diversity are needed.

Figures 3 & 4: Combine a panels into a single figure and move b panels to supplement.

Line 197-200: Why might mammals and/or a global scale analysis produce different results?

Line 210-211: Not all taxa follow this (e.g. marine fishes – see work by Rabosky et al., Title et al.) Perhaps note that a further text would be to see if opposite result occurs in taxa where speciation rates are higher at the poles.

Line 227: Clarify timeframe of "millennia".

Line 338: There is no justification for this measure or citations of its use elsewhere.

Line 521: "Note...." This is an important result!

Line 537: Why $\Delta AIC > 5$? Justification needed

Tables specifying collinearity among predictor variables are needed in the supplement.

Dear editor, reviewers,

We are grateful for the very constructive comments on our manuscript. We've invested significant time and effort into addressing all issues raised by the three reviewers and we now feel confident that the manuscript is very novel, with the primary results being articulated alongside existing theories on global biodiversity patterns. Below, you will find our responses to each of the reviewers' comments (our responses are preceded by an asterisk and are in bold) with new line numbers where appropriate. Additions and modifications in the revised version are highlighted with red color. We further attach our responses to the reviewers' comments (Reviewer 1 and 2) in the marked version of the paper in two separate word files.

Reviewers' comments:

Reviewer #1 (Remarks to the Author):

Review : « Spatiotemporal drivers of genetic diversity in terrestrial mammals »

I congratulate the authors for the effort of assembling such large datasets. The novelty in their study is to consider phylogenetic diversity, past climatic and land use changed at global scale. This is an important topic to better understand the pattern of genetic diversity at global scale, a level of biodiversity that just start to be considered. I appreciate the effort done by the author. However, I think that in its current form, the paper is mainly descriptive, and requires strong clarifications on the statistical analysis. We are now needed more than a descriptive papers in the field of global spatial and temporal patterns of genetic diversity after the seminal papers of Miraldo et al (2016) and Millette et al. (2019). If the authors are able to relate more directly their results to theory and assumptions, and to clarify, justify or modify their statistical analyses, the paper can have potential to be published. But it requires a major revision and clarifications about the usefulness of the predictions. But if the authors are not able to clarify the utility of the predictions (Fig 3 and 4), I am effraid that the study lack of original results to be published in Nature Communications. I summarize below my main comments. Most of my comments can be found in a marked version of the manuscript.

*** We thank the reviewer for the detailed and constructive criticism. We carefully followed all of their suggestions including, better articulating the originality and novelty of our study both in terms of the tested hypotheses and in terms of data. Moreover, we now better clarify/justify our statistical approaches and the utility of our predictions. Below, we provide our detailed responses.**

-My first point is related to the need to clarify the novelty of the paper in regards of the two published papers of Miraldo et al. (2016) and Millette et al. (2019) that also included mammals? Why their database is largest since the number of sequences for mammals seems to be smaller (see my comment in the text). I assume that it has been updated in the time, and for Millette, only COI is available, but it needs to be clearly explained. This point is related to the need of a clear position of the current work in regards on previous work on the topics, ie recent state of the art on the global patterns on genetic diversity in mammals from at least this two papers.

*** We agree with the reviewer and now better justify the novelty of our study, particularly with regards to hypothesis testing and data utilization.**

Regarding novelty: The two studies highlighted by the reviewer (i.e. Miraldo et al. 2016 and Millette et al. 2019) inferred the existence of a latitudinal gradient in the genetic diversity of terrestrial mammals, at least at a coarse (latitudinal band) scale. They did not, however, test for the controlling role of evolution and/or climate on spatial patterns of genetic diversity across the globe. This is in part due to a lack of adequate data and the different focus of these studies (e.g. Millette et al. 2019 on human impacts). In contrast, our study utilised much more data (see below) to assesses long-standing hypotheses on the role of evolutionary history, climatic fluctuations since the last Glacial Maximum and long-term and more recent human impacts on current-day patterns of interspecific diversity. We now clarify the gap of knowledge and the novelty of our study in lines 7-12, 36-45, 72-74, 89-90, 236-239, 279-283.

Regarding data availability: Our extended dataset includes almost double the amount of sequences for cytb (24,395 sequences, ~85% increase) compared to those used by Miraldo et al. 2016, while for co1 the increase is relatively smaller (22,570, ~15% increase). Note that Miraldo et al. in their paper report the raw and not the utilized (i.e. filtered) georeferenced sequences. Likewise, the number of georeferenced co1 data used in our study has > 5,000 sequences more compared to Millette et al. (16,890), while Millette et al. failed to use the very informative cytb marker. We've added this information in the main text to clarify the novelty of our study in terms of data (lines 92-96)

-A second point is related to the assumptions in the introduction : they are very general. Is it possible to provide more specific assumptions for mammals ? Why having chosen to focus on mammals ? Miraldo et al. included amphibians and Millette et al invertebrates, birds and fishes : they focus on vertebrates. Is there any specific reasons? Probably that the conclusions can be more general if extended to birds and amphibian (ie terrestrial vertebrates) at least. Looking for mammals is fine but it requires some justification and specific assumptions. Related to assumptions, I am expecting a specific assumption for each factor tested. This can help to give a direction to the paper.

Our choice to focus on mammals is primarily based on the spatial and taxonomic coverage of available georeferenced data for mammals, which is much higher and more informative compared to any other animal group. Furthermore, the decision reflects the availability of extensive taxonomic and systematic information (i.e. taxonomy, distributions, phylogenies) for this animal class, a consideration needed to make robust inferences regarding different hypothesised drivers of genetic diversity (lines 90-92).

We see the reviewer's point that our a priori hypotheses need to be better communicated. We now clearly describe how we test the importance of:

- 1. Two major and complementary evolutionary hypotheses, i.e. the "evolutionary speed" and the "Red Queen" hypotheses, that specifically predict positive covariation between intraspecific genetic diversity and interspecific biodiversity (i.e. species richness and phylogenetic diversity). These hypotheses were formulated to explain global biodiversity patterns across organisms and do not specifically refer to mammals. To the best of our knowledge, there's no specific hypotheses on global biodiversity gradients that is formulated for mammals alone. However, the original formulation of the "evolutionary speed" hypotheses was based on ectotherms, providing this study of**

mammal genetic diversity with the opportunity to explore the relevance of this hypotheses for endotherms. Detailed description of these two hypotheses, as well as their predictions and previous results are now given in lines 57-69, 225-243.

- 2. The role of abrupt climate change events during the most recent deglaciation period in shaping the global pattern of genetic diversity and provide specific expectations stemming from the climate stability hypotheses in lines 76-83 (see also lines 250-266 for a discussion of our results).**
- 3. The effect of long-term and more recent human land modification and how this could potentially influence the amount of genetic diversity in mammal species assemblages globally (see lines 83-87 for expectations and lines 268-276 for discussion on data limitations and lack of signal).**

-Concerning the main figures and results:

- In figure 1, I feel that it is not necessary to show the outputs for the both markers since the trends is similar. I suggest to keep the more robust (cytb seems to have a better coverage).

*** Indeed, cytb has a much better spatial and taxonomic coverage compared to co1. However, since there's little spatial and taxonomic overlap between the two genetic markers, we believe that the results obtained with co1 are crucial both in confirming the inferences based on cytb, and in enhancing the contribution of climate stability in the global projections of GD. Therefore, in accordance with the suggestions of Reviewer 3 (see below) and the broad usage of the co1 gene in evolutionary and macroecological studies (e.g. Millete et al. 2019; Manel et al. 2020), we think that the results of this marker should remain in the main figures.**

- The Figure 2 is a sensitivity analysis. So it can be moved in supplementary material: it helps to discuss the power of the main analysis but do not bring any new result.

*** Figure 2 is indeed a sensitivity analysis, which is crucial for interpreting many aspects of our manuscript.**

- 1. It exposes the gap of knowledge and the Wallacean shortfall for genetic diversity globally and indicates highly-diverse regions of the world without adequate genetic information.**
- 2. It suggests analytical ways for reducing the noise stemming from the spatially and taxonomically fragmented nature of genetic data across organisms. It further increases the transparency of our study in terms of utilized data.**
- 3. It shows the significant statistical association between genetic diversity and species richness and phylogenetic diversity, thus helping the reader understand where the predictive maps in new Figure 3 stem from.**

- But more important, is my misunderstanding of the results of the Figure 3 and 4. Probably that I miss something but I do not understand the utility of the predictions. I do not understand why not analyzing directly the relation between genetic diversity and latitude as it has already been done by Miraldo et al for mammals in 2016 (Fig. 3A) ? Miraldo et al showed that GD was lower at high altitude and higher in the tropics. I feel that the information the authors are looking for in the

predictions are already available in the data. But again, it is possible that I do not understand this analysis and its background, and in this case, it requires more explanations.

*** The revised version of this manuscript now directly justifies the utility of our predictive models. See lines 14-16, 48-51, 203-206, 220-223, 278-292, 301-309.**

As we already mentioned above, the significant lack of data in previous studies (i.e. Miraldo et al. 2016 and Millette et al. 2019) allowed for only inferences regarding genetic diversity at a coarse latitudinal band scale. Conversely, our models provide fine-scale resolution prediction of genetic diversity globally. In doing so they help to fill major information gaps and reduce the so called “Wallacean shortfall” for genetic diversity, enhancing our ability to test theories (basic science) and improve biodiversity conservation (applied science):

i) The predicted global distribution of GD at a grid-cell spatial scale will significantly enhance our capacity to mechanistically model (that is, simulate) the effects of important evolutionary processes (including mutation, drift, gene flow and natural selection) on global biodiversity gradients (lines 283-286).

ii) As genome-wide data becomes available at finer spatial resolutions, our predictive maps will further serve as a baseline for assessing the role of the above evolutionary processes in driving local and regional deviations (studies cited in lines 40-45) from the predicted global patterns of GD (lines 286-292)

iii) Identifying areas of particular importance for primary dimensions of biodiversity will help us make informed decisions that will largely contribute to meeting major conservation targets for halting the accelerating biodiversity loss (e.g. highest genetic diversity is concentrated in the tropical regions that are the most exposed to global change; lines 301-309).

Furthermore, and following the suggestion of Reviewer 3, we have excluded the analysis at the latitudinal band scale, as the results at this scale were less relevant for the message we are trying to convey. We believe that the exclusion of these results will help the reader better understand the novel contribution of our finer-scale global map and analysis of GD. Additionally, and following the suggestion of Reviewer 3, we have merged the predictive maps for the two genetic markers (previous Figs 3 and 4) into one figure (Fig. 3) and moved the spatial distribution of model residuals to the supplementary information (Supplementary Fig. 4).

-I have a quick look on the discussion but as it can strongly change, I did not work more on it. The discussion should not repeat the introduction and should interpret the results.

*** We have overhauled the discussion so it is more orientated on the hypotheses we tested- and on the interpretation and significance of our results**

Finally, I have questions on the statistical method (model):

1) In the linear regression, I am not sure whether the variable GD has been transformed or not before applying linear regression. It is probably not a normal variable. Did the authors check for its distribution ? The linear regression on the raw variables is not appropriated. I potentially understand that a square root transformation has been applied in the grid cell ? Is it true ? Why not at the other

spatial scales? Explanations on this point are essential since this is the starting point of the analysis. It needs clarifications in the paper.

*** As the distribution of GD at the grid cell scale was highly skewed towards zero, we transformed GD to normality using its square root. All subsequent statistical analysis at the grid cell scale were based on the transformed GD. We now clarify the applied transformation at the grid cell scale in lines 471-475.**

2) The authors tested all models and they didn't first check for variables/factors colinearity but they removed models with correlated variables. I am wondering how many models were removed ? I am wondering why not first checking for colinearity and then applying models on uncorrelated variables : it seems a more classical strategy. I assume that variables are highly correlated. Which one? I really need more information on this point, and how correlations do not confound the results.

*** We thank the reviewer for pointing out the lack of information on multicollinearity among the eight independent variables. This was also pointed out by Reviewer 3. We specifically chose to use the two alternative statistical methods (that is, hierarchical partitioning and multimodel inference) because they offer some advantages over more classical approaches in dealing with multicollinearity (e.g. evaluate the contribution of both SR and PD jointly and not excluding one or the other as highly collinear). We justify the use of our two alternative statistical approaches and indicate where multicollinearity may confound our results in lines 504-506, 525-529. As suggested by Reviewer 3, we further include two tables with pairwise correlations among all independent variables both at the grid cell and zoogeographic regions scales (Table S1 and Table S2).**

For a more detailed exploration of spatial and statistical distributions of the variables, as well as the relationships between independent and dependant variables, the reviewer can visit the interactive version of our manuscript (<http://geneticgeography.com>; see Supplementary File S2 for instructions on how to use the web application).

Yet, all the explanatory variables are measured in different units : if analyzed together, they should probably be normalized?

*** There's no specific requirement for linear regression (that is, Ordinary Least Squares estimation) to have the distribution and units of measurements of independent variables normalized.**

3) The test of spatial autocorrelation indicated no spatial correlation in their analysis. Can the authors explain more what does this spatial autocorrelation reflect and how it is different from the one discussed in the paper of Gratton et al (2017). Why it is not useful to add a term of spatial autocorrelation in the model?

*** We provide justification of why spatial autocorrelation in residuals may be problematic in statistical inference in lines 546-550. Spatial autocorrelation in residuals should be accounted for only when it is detected, which is not the case in our models. Moreover, Gratton et al (2017) criticizes the inferences at the latitudinal band scale, due to potential challenges arising from spatial autocorrelation, which we have addressed in our analysis.**

Finally, I recommend to avoid to use too many abbreviations. The paper becomes difficult to read.

*** We did our best to remove as many abbreviations as possible from the main text.**

I was not able to evaluate the sections related to the description of climate stability and Human footprint since I am not a specialist of this field, as well as of phylogenic diversity. I observed that the two sections in the method describing this part were really more detailed than the other one. I am wondering if those sections were not too detailed or the other methods sections not enough : probably need to find a compromise between both.

*** Since the paper that describes the methods and data for climate stability is now published in Nature Climate Change, we significantly reduced the size of the respective part and we now refer the user to the published paper. This helped us in significantly reducing the overall size of the methods, as well as the number of citations. Additionally, and following the suggestion of Reviewer 2, we slightly lengthened the part where we describe the estimation of species richness and phylogenetic diversity. The three sections are now more equally distributed in the methods (lines 403-468).**

I included more comments in a marked version of the paper.

Suggestion of additional biblio related to the topic

Tucker C, Cadotte M. Unifying measures of biodiversity: Understanding when richness and phylogenetic diversity should be congruent. *Diversity and Distributions* 19, 845-854 (2013)
Voskamp A, Baker DJ, Stephens PA, Valdes PJ, Willis SG. Global patterns in the divergence between phylogenetic diversity and species richness in terrestrial birds. *Journal of Biogeography* 44, 709-721 (2017).

*** We have added the suggested references in the main text (line 34)**

[SEE ADDITIONAL COMMENTS IN ATTACHED DOCUMENT]

Reviewer #2 (Remarks to the Author):

In the manuscript #NCOMMS-19-37347, the authors evaluated >50,000 mitochondrial genetic sequences for >1,500 mammal species to assess the biodiversity, climatic and anthropogenic correlates of genetic diversity worldwide. Moreover, the authors also challenge the Wallacean shortfall for the most basal level of biological diversity. Based on wide databases and sophisticated methods, this study presents an innovative piece of work in an elegant theoretical context. I have just a few comments and suggestions (please, see attached document), which I hope to help the authors improve just details.

Sincerely,

Matheus S. Lima-Ribeiro

*** We are very grateful to the reviewer for his positive assessment of our manuscript and helpful suggestions. We tried to address all of his comments (see the attached marked word file)**

[SEE ADDITIONAL COMMENTS IN ATTACHED DOCUMENT]

Reviewer #3 (Remarks to the Author):

Review of “Spatiotemporal drivers of genetic diversity in terrestrial mammals” by Theodoridis et al.

This manuscript addressing the relationship between genetic diversity based on two genes, phylogenetic diversity, and species richness in mammals globally while also taking into account variation in historical climate since the LGM and both past and present human impacts. The question is interesting and timely, the dataset and analyses are laudable, and the writing is well done. My concerns are as follows:

1) The manuscript lacks discussion of or justification for why *cytb* and *co1* were selected, what their significance is, and the extent to which they are representative of genetic diversity more broadly

*** This is a valid point and something we have followed up-on. The revised manuscript now provides a detailed justification of the use of the two selected mitochondrial markers.**

Our choice of *cytb* and *co1* was primarily based on data available, noting that we need associated spatial information for genetic samples analysed. These two genes have been extensively used in taxonomic, phylogenetic and phylogeographic studies because their intraspecific variation is a good indicator of processes, such as population divergence and extirpation (due to properties such as e.g. rapid evolution, homoplasy, and low recombination). Therefore, it was no surprise that they were found to be the most appropriate genes for the purposes of our study. We justify the choice of these markers and acknowledge their limitation in lines 96-100, 286-289.

2) The manuscript lack justification for why analyses were conducted using 4 degree latitudinal bands. Biologically, this categorization seems odd given the longitudinal variation within and among continents and the level of environmental heterogeneity with longitude. The most biologically relevant of the three spatial scales of analysis is the zoogeographic regions, but these show the “weakest” results, or at least results that differ from the grid cell and 4 degree latitude bands. Discussion of the results minimizes the geographic regions. I would suggest that the 4 degree band analysis be excluded and the comparison between the grid cell and zoogeographic regions be given more weight. In this context, the relationship between genetic diversity and species richness is much weaker.

*** We agree with the reviewer on the utility of the analyses at the latitudinal band scale. We therefore excluded the latitudinal band scale from our analyses. We now focus on the utility of inferences and predictions at the grid cell scale (see responses to Reviewer 1), and further discuss our choice of the zoogeographic regions scale and the respective results in lines 101-104, 112, 122-124, 157-163, 185-188, 256-263.**

3) The manuscript is framed largely in a conservation biology context with a secondary focus on the basic evolutionary question at hand. No alternative hypotheses for a positive relationship between genetic diversity and phylo/richness are presented or explored, which essentially sets to paper up as

confirmatory science rather than as a test of the relationship between genetic diversity and phylo/richness. More discussion of conflicting results in the literature and the ways in which this analyses does (or does not) resolve them is needed, particularly given the fact that mammals may differ from other taxa in which differing results have previously been found.

*** Following also the suggestions of Reviewer 1, we significantly modified our manuscript and put our results into context with two major hypotheses that predict positive covariation between intraspecific genetic diversity and interspecific diversity, i.e. species richness and phylogenetic diversity. Detailed description of these two hypotheses, as well as their predictions and previous results are now given in lines 53-90, 225-276 (see also our responses to Reviewer 1).**

Minor comments:

Line 12: Clarify that the covariation is positive.

*** Done**

Line 15: Replace “the” information gap with “an” information gap and clarify that the most basal level of biological diversity is genetic (rather than taxonomic, for example).

*** Done**

Line 27-28: Tell us more about the opposing evidence and what might explain differences

*** We have restructured the Introduction (see lines 40-45). We do not elaborate more on the results of regional studies as they are irrelevant and to the hypotheses we are testing at global scale. However, we now provide discussion on the potential role variation in evolutionary processes, such mutation, drift, gene flow and natural selection, across space and taxa in explaining the lack of consistent signal at these scales (lines 286-292)**

Line 43: long-standing theories, such as??

*** We have restructured the introduction (see lines 53-74)**

Line 51-54: This would be a place to add alternatives

*** Done (see lines 53-74)**

Line 71-72: Why these two?

*** See response above on the choice of markers**

Line 114: Why the square root?

*** See lines 471-475 for justification (see also the relevant response to Reviewer 1)**

Line 125-126: Consider adding the spatial distribution of these as a supplement to assess how overall results may be affected by geographic sampling biases

*** Done, see lines 161-163 and Supplementary Fig. 1**

Line 129: It's unclear what the biological significance of the latitudinal bands is.

*** Latitudinal bands are now excluded**

Line 137: What happens if you limit the geographic extent to only the well represented ones? Biologically this is the most relevant measure.

*** We could not apply filtering at the zoogeographic region scale as the spatial and taxonomic coverage at this scale and for this marker (co1) is very unbalanced (see also Supplementary Fig. 1)**

Line 141: Rephrase section head.

*** Done. It now reads “Contributions of climate stability to global genetic diversity”**

Figure 1: Add sentence to main text defining/justifying focus on IC and SAW and which is a better measure (b/c results differ, particularly for importance of species richness)

*** Done. See lines 131-132, 138-141**

Figure 2: Axes labels for genetic diversity and phylogenetic diversity are needed.

*** It is not clear what the reviewer asks. In our version of the manuscript axes labels in the scatter plot are present.**

Figures 3 & 4: Combine a panels into a single figure and move b panels to supplement.

*** Done. See new Figure 3**

Line 197-200: Why might mammals and/or a global scale analysis produce different results?

*** We have restructured the Discussion. The results of regional studies are now present only in the Introduction (lines 41-43) and some discussion on the discrepancies in lines 286-292.**

Line 210-211: Not all taxa follow this (e.g. marine fishes – see work by Rabosky et al., Title et al.) Perhaps note that a further text would be to see if opposite result occurs in taxa where speciation rates are higher at the poles.

*** See lines 228-233**

Line 227: Clarify timeframe of “millennia”.

* **Done. See line 256**

Line 338: There is no justification for this measure or citations of its use elsewhere.

* **Latitudinal bands are now excluded**

Line 521: “Note....” This is an important result!

* **See comment above on the use of zoogeographic regions and also Supplementary Fig. 1**

Line 537: Why delta AIC > 5? Justification needed

* **Done. See line 532-534**

Tables specifying collinearity among predictor variables are needed in the supplement.

* **Done. See Tables S1 and S2.**

Reviewers' Comments:

Reviewer #1:

Remarks to the Author:

Revision : "Spatiotemporal drivers of genetic diversity in terrestrial mammals"

Compared to the first submission, the manuscript has certainly improved. The major novelty of the study is to show an effect of past climate change on genetic diversity. I agree that having two markers, and two statistical approaches to test assumptions is a robust approach and I like it (Figure 1). I am largely satisfied with the author's reply to my previous statistical concerns, and I am glad that the authors use some of my suggestions and those of the other referees.

However, I still have some minor concerns related to the statistical approach, and to the lack of a deeper interpretation of the result to discuss the processes that generate the observed patterns. Those concerns make the manuscript not ready yet for publication in a high-profile journal such as Nature Communications. I remark that I regard this as a valuable study which does deserve publication, but I urge the authors to make a further effort to provide potential readers with a consequential exposition of their work and its significance for macroecological theory. I hope my comments may help a bit.

A more detailed discussion of the assumptions and how results of the study are used to accept or reject the assumptions are clearly needed. The current version of the paper lacks of an interpretation of the results related to potential hypothesis at the origin of the detected patterns. In brief I am expecting in a revision to have more specific discussion on behind processes related to the detected patterns, in relation with the results of this study and not just a general discussion of the potential theory. The current version the paper is very descriptive, while we expect more when we start reading the introduction (l48-49: "the global pattern of ... biodiversity is vital both for assessing the underlying processes shaping" or l72: "provides unique opportunities both in assessing long-standing theories..."). I feel the database collected and the outputs from this study have great potential to discuss standing theories and It would be a shame not to go further with all the database and analysis work that has already been done.

(1) I appreciate that the authors have made an effort to develop assumptions to explain the observed patterns in the introduction. However, their results are not really interpreted and discussed in regards of these assumptions. I would like to have a specific description in the introduction (and eventually in more details in the MM) of the expected relations between GD and the historic variables. Yet I would recommend to add a supplementary material with slope's values of the regression (ie. full model outputs) since this is an essential component of the biological interpretation of regression. It can help the authors to go deeper in the discussion of the results, as well to support or reject assumption/expectation. The stability hypothesis assumption is used in the discussion to explain the negative correlation between the past climatic temperature trends and genetic diversity. I would like to see more explanations in the method on the influence of each predictor on genetic diversity: what are the expectations on the trend and the variability of the temperatures of past climate change variables? What happen if the trend is negative? Do the authors consider the absolute value of the trend? Or the trend is always positive in the considered period? Are there expected differences between the interpretation of the two variables trends and variability? Same for precipitation. In summary, I do not find very clear the interpretation of the climatic temperature trends and variability and that a negative correlation is related to climate stability. This should be better explained in the paper.

(2) I think that alternative assumptions to evolutionary speed and red queen assumptions can be discussed to interpret outputs. For example, it is not possible to interpret past precipitation changes in relation with productivity changes, ie energy assumption? And how it can influence the genetic diversity?

(3) In the discussion on the hypothesis of climatic stability, I would suggest adding a discussion on demographic scenarios that can be associated to past climatic changes as species contraction and range expansion, since they are major reasons to explain variations in genetic diversity.

(4) I think that an analysis of the spatial autocorrelation in the raw data of GD is missing. I would suggest to add a spatial auto-correlogram of the raw data, and, depending of the results (presence of spatial autocorrelation among grid cells) probably that the Pearson correlation is not adapted to test the correlation between GD and SP or PD. As these spatial variables are observed over the same locations, and play a similar role (no variable to explain), I suggest to use the modified t-test of spatial association (Clifford, P., Richardson, S. & Hemon, D. Assessing the significance of the correlation between two spatial processes. *Biometrics* 45, 123–134 (1989)). I am sorry to miss this point in my first reading.

I am fine with the regression between GD and other environmental variables since no spatial autocorrelation in the residuals have been detected.

(5) Related to the predictions of genetic diversity, I thank you the authors for additional explanations agree that predictions are needed and that they can help to reduce the "Wallacean shortfall", but they cannot reduce alone the shortfall. In the current study (i) the highest explanatory power in the model is about 0.4. It means that important explanatory variables are missing to explain the GD and, (ii) gaps are still present in some continents (eg. Africa) and only few points were used to train the model (Fig. 3). For those reasons, the model used to make the predictions is far from being perfect. I would recommend to discuss the limits of the model, and to change a little the conclusion in some section of the manuscript, to say that predictions can help to reduce the Wallacean shortfall, but they will not alone reduce the Wallacean shortfall, and probably the most important task is to sample more data, mostly in the gaps.

(6) I also would like to see a more detailed description of the results of the sampling gaps in mammals for GD: in which countries/continents are the highest gaps?

(7) I would recommend to have a more appealing title.

I was not able to check the data since the link is yet not public: <http://geneticgeography.com>.

More detailed comments

L 30-31: "However, the extent to which GD covaries with interspecific diversity remains unclear" Do not fully agree is still debated. It is controversial depending on the sampling, the case study..

L30-31: "long-standing theories" I would say "existing"

L34: "considering the potential effects of Late Quaternary climate change and anthropogenic land-use" : unclear if the authors also test current climate effect?

L36-37: It is important to be more specific: how is this effect: positive or negative?

L36 to 39: I am waiting here a deeper interpretation of the results.

L37-39 I suggest to be more cautious with the predictions and their impact (see my general

comment")

L48. I suggest to add description or knowledge in the sentence "the global pattern"◊The description of the global pattern or the knowledge of

L53: to explain and link: unclear

L57 to 61: too long sentence

L74 "have not been done"◊is missing

L99 to 110: I would suggest here to mention the colonization hypothesis and to relate past climate changes to demographic events as contraction and range expansion that can explain variation in genetic diversity.

L135-136. If possible, I would like to have more informations/results on the trends between GD and climatic variables.

L151: possible to add "positively" covaries ...

L163: is indicative of◊reflects

L181: I do not understand what is a secondary contribution?

L195-196: I am waiting more information on the uncomplete sampling of GD across the global: where are the gaps? Where do future research need to put their effort?

L202. "higher temperature stability is associated with higher values of mammalian GD"◊I probably miss something but in Figure 1, I see a negative relationship between GD and any temperature variables (trends and variability).

L201: "plays a significant role in shaping the distribution of GD " ◊not possible to give more explanation of which role : it can help to clarify your main message of the paper.

L201 to 208 is only one long sentence and is very difficult to read.

L242-243. I suggest to better characterize the effect of past climate change on GD.

L245-246: need to be rewritten considering that predictions should be interpreted more caution.

L248-252: Unclear. Evolutionary assumption is not the only one to explain micro-macro continuum. It is like the evolutionary hypothesis was assimilated to the only theory linking micro- to macro evolution (ie correlation among GD and species diversity). Yet there are a panel of assumptions and it is important to mention them and to see how results from empirical study can support them or not them.

L257. Evolutionary assumption◊ micro-macro continuum (and perhaps also for l251)

L259: The study on fish shows that that the evolutionary speed hypothesis is not the only one to explain the correlation GD-SD.

L273-274. Unclear. a positive relationship between GD and climate stability has been detected since climate stability is negatively correlated to temperature trends and variability ◊what is the effect on extinction and diversification?

L445: I feel that here we miss the information of how climate stability is related to the trend and the variability of temperature (and precipitation). From the results I assume that both trends and variability are negatively correlated to stability (?) The information comes later in the discussion. Anyway, I recommend to add more information to facilitate the interpretation of the detected relations between GD and past climatic variables, as well as assumptions / expectation.

Reviewer #2:

Remarks to the Author:

The authors answered all the points I raised earlier. Thank you very much for considering my comments. I have no comments on this version.

Sincerely

Matheus lima-Ribeiro

Dear editor, reviewers,

We are grateful for the additional minor comments on our manuscript. We've addressed all issues raised by the Reviewer 1 and we now feel confident that the manuscript is ready for publication, as also suggested by Reviewer 2. Below, you will find our responses to each of the reviewer's comments (our responses are preceded by an asterisk and are in bold) with new line numbers where appropriate. Additions and modifications in the revised version of the manuscript are highlighted with red color.

Reviewer #1 (Remarks to the Author):

Compared to the first submission, the manuscript has certainly improved. The major novelty of the study is to show an effect of past climate change on genetic diversity. I agree that having two markers, and two statistical approaches to test assumptions is a robust approach and I like it (Figure 1). I am largely satisfied with the author's reply to my previous statistical concerns, and I am glad that the authors use some of my suggestions and those of the other referees.

However, I still have some minor concerns related to the statistical approach, and to the lack of a deeper interpretation of the result to discuss the processes that generate the observed patterns. Those concerns make the manuscript not ready yet for publication in a high-profile journal such as Nature Communications. I remark that I regard this as a valuable study which does deserve publication, but I urge the authors to make a further effort to provide potential readers with a consequential exposition of their work and its significance for macroecological theory. I hope my comments may help a bit.

A more detailed discussion of the assumptions and how results of the study are used to accept or reject the assumptions are clearly needed. The current version of the paper lacks of an interpretation of the results related to potential hypothesis at the origin of the detected patterns. In brief I am expecting in a revision to have more specific discussion on behind processes related to the detected patterns, in relation with the results of this study and not just a general discussion of the potential theory. The current version the paper is very descriptive, while we expect more when we start reading the introduction (148-49: "the global pattern of ... biodiversity is vital both for assessing the underlying processes shaping" or 172: "provides unique opportunities both in assessing long-standing theories..."). I feel the database collected and the outputs from this study have great potential to discuss standing theories and It would be a shame not to go further with all the database and analysis work that has already been done.

*** We thank the reviewer for their additional detailed minor comments/suggestions that helped us to further improve our manuscript. We now provide more specific expectations and a detailed discussion with regards to the relationships between GD and SR/PD (see also the very recent Ref. 30 for a review), and the relationships between the alternative definitions of climate stability and GD. Additionally, we addressed all of the reviewer's methodological concerns. Below you can find our specific responses.**

(1) I appreciate that the authors have made an effort to develop assumptions to explain the observed patterns in the introduction. However, their results are not really interpreted and discussed in regards of these assumptions. I would like to have a specific description in the introduction (and eventually in more details in the MM) of the expected relations between GD and the historic

variables. Yet I would recommend to add a supplementary material with slope's values of the regression (ie. full model outputs) since this is an essential component of the biological interpretation of regression. It can help the authors to go deeper in the discussion of the results, as well to support or reject assumption/expectation. The stability hypothesis assumption is used in the discussion to explain the negative correlation between the past climatic temperature trends and genetic diversity. I would like to see more explanations in the method on the influence of each predictor on genetic diversity: what are the expectations on the trend and the variability of the temperatures of past climate change variables? What happen if the trend is negative? Do the authors consider the absolute value of the trend? Or the trend is always positive in the considered period? Are there expected differences between the interpretation of the two variables trends and variability? Same for precipitation. In summary, I do not find very clear the interpretation of the climatic temperature trends and variability and that a negative correlation is related to climate stability. This should be better explained in the paper.

*** We now provide more specific expectations regarding relationships between intra- and interspecific biodiversity under our proposed hypotheses (Lines 52-80).**

We further clarified the definition of climate stability (i.e., trend and variability), the expected relationships with GD under each definition, and the interpretation of our results (lines 82-94, 192-210, 286-306, 470-477)

Moreover, as suggested by the reviewer, we have added a Supplementary Table (S6) reporting the slope/coefficient for each explanatory variable across all retained models.

(2) I think that alternative assumptions to evolutionary speed and red queen assumptions can be discussed to interpret outputs. For example, it is not possible to interpret past precipitation changes in relation with productivity changes, ie energy assumption? And how it can influence the genetic diversity?

*** We have now added one more major hypothesis (the “time and area” hypothesis) that predicts a positive covariation between GD and interspecific biodiversity. We further discuss the intermediate role of energy-driven productivity, as integrated in recent interpretations of the “evolutionary speed” hypothesis (see Ref. 33), in promoting overall biodiversity in the tropics. Our paper now focuses on three major and basal theories (that is, the “evolutionary speed”, the “time and area” and the “Red Queen” hypotheses) that make specific predictions regarding the covariation between GD and SR / PD. These theories further constitute the basis for most eco-evolutionary theories that have been invoked to link intra- and interspecific biodiversity at global scale (see also Ref. 30). See lines 52-80 for an Introduction in these theories and 249-284 for a Discussion and interpretation of our results. Regarding the precipitation changes, and as suggested by the Reviewer, their effects in current GD patterns are only discussed in relation to demographic scenarios, i.e, contraction and expansion of biomes driving adaptive divergence, local extinction of populations etc (See also comment below).**

(3) In the discussion on the hypothesis of climatic stability, I would suggest adding a discussion on demographic scenarios that can be associated to past climatic changes as species contraction and range expansion, since they are major reasons to explain variations in genetic diversity.

*** We now refer to demographic processes with regards to climate stability both in the Introduction (lines 82-94) and in the Discussion (286-306).**

(4) I think that an analysis of the spatial autocorrelation in the raw data of GD is missing. I would suggest to add a spatial auto-correlogram of the raw data, and, depending of the results (presence of spatial autocorrelation among grid cells) probably that the Pearson correlation is not adapted to test the correlation between GD and SP or PD. As these spatial variables are observed over the same locations, and play a similar role (no variable to explain), I suggest to use the modified t-test of spatial association (Clifford, P., Richardson, S. & Hemon, D. Assessing the significance of the correlation between two spatial processes. *Biometrics* 45, 123–134 (1989)). I am sorry to miss this point in my first reading.

I am fine with the regression between GD and other environmental variables since no spatial autocorrelation in the residuals have been detected.

*** Following the reviewer's suggestion, we applied the modified t-test for further testing the significance of spatial association between GD and PD and SR and we replaced all P-values in the respective sections of Results (lines 150-154). As expected, the correlation values (modified t-tests) are the same as in Fig. 2, and the majority of modified tests are highly significant across markers and data subsets. The method is described in lines 549-554 and full results are given in Tables S3 and S4 and further mentioned in Fig. 2 (caption). We do not present the results of spatial autocorrelation in the raw data of GD (spatial auto-correlogram) due to the high complexity of the results when using multiple combinations of minimum taxonomic thresholds (see Fig. 2) and distance classes.**

(5) Related to the predictions of genetic diversity, I thank you the authors for additional explanations agree that predictions are needed and that they can help to reduce the “Wallacean shortfall”, but they cannot reduce alone the shortfall. In the current study (i) the highest explanatory power in the model is about 0.4. It means that important explanatory variables are missing to explain the GD and, (ii) gaps are still present in some continents (eg. Africa) and only few points were used to train the model (Fig. 3). For those reasons, the model used to make the predictions is far from being perfect. I would recommend to discuss the limits of the model, and to change a little the conclusion in some section of the manuscript, to say that predictions can help to reduce the Wallacean shortfall, but they will not alone reduce the Wallacean shortfall, and probably the most important task is to sample more data, mostly in the gaps.

*** Following the Reviewer's recommendations, we now discuss the limitation of our models, modified our conclusions in several places of the manuscript, and highlight the need for more field-work to enhance, complement and validate global models (lines 14-16, 221-223, 243-247, 318-333, 348, 350).**

(6) I also would like to see a more detailed description of the results of the sampling gaps in mammals for GD: in which countries/continents are the highest gaps?

*** We now provide global maps of sequence availability and taxonomic coverage for both markers in Supplementary Figures 1 and 2. We refer to these figures in several places in the manuscript (e.g. lines 190, 319).**

(7) I would recommend to have a more appealing title.

*** We agree with the reviewer that a more appealing and result-driven title is necessary. We changed the title to “Evolutionary history and past climate change shape the global distribution of genetic diversity in terrestrial mammals”**

I was not able to check the data since the link is yet not public: <http://geneticgeography.com>.

*** We apologize to the reviewer for not including the credentials in the rebuttal letter. This information was previously provided in the “Instructions for geneticgeography.com” file. The reviewer can access the web application by using the following credentials:**

Username: geneticdiversity

Password: geneticdiversity

More detailed comments

L 30-31: “However, the extent to which GD covaries with interspecific diversity remains unclear”
Do not fully agree—is still debated. It is controversial depending on the sampling, the case study.

*** We modified the respective part**

L30-31: “long-standing theories”→I would say “existing”

*** Done**

L34: “considering the potential effects of Late Quaternary climate change and anthropogenic land-use” : unclear if the authors also test current climate effect?

*** We replaced “Late Quaternary” with “past”**

L36-37: It is important to be more specific: how is this effect: positive or negative?

*** Done**

L36 to 39: I am waiting here a deeper interpretation of the results.

*** Done**

L37-39→I suggest to be more cautious with the predictions and their impact (see my general comment”)

*** Done. See our response above.**

L48. I suggest to add description or knowledge in the sentence “the global pattern”→The description of the global pattern or the knowledge of

*** Done**

L53: to explain and link: unclear

*** We have reformulated the sentence and is now more specific**

L57 to 61: too long sentence

*** We have split the sentence in three parts**

L74 “have not been done” is missing

*** Done**

L99 to 110: I would suggest here to mention the colonization hypothesis and to relate past climate changes to demographic events as contraction and range expansion that can explain variation in genetic diversity.

*** Done. See also our response above.**

L135-136. If possible, I would like to have more informations/results on the trends between GD and climatic variables.

*** Done**

L151: possible to add “positively” covaries ...

*** Done**

L163: is indicative of reflects

*** Done**

L181: I do not understand what is a secondary contribution?

*** We removed “secondary”**

L195-196: I am waiting more information on the uncomplete sampling of GD across the global: where are the gaps? Where do future research need to put their effort?

*** We have added comment in lines 188-190 and 328-331 and a reference to the new Supplementary Figures 1 and 2.**

L202. “higher temperature stability is associated with higher values of mammalian GD” I probably miss something but in Figure 1, I see a negative relationship between GD and any temperature variables (trends and variability).

*** We now better clarify the definition of climate stability in terms of trend and variability. See our response above and new lines 193-210.**

L201: “plays a significant role in shaping the distribution of GD “ ànot possible to give more explanation of which role : it can help to clarify your main message of the paper.

*** We now better clarify the definition of climate stability in terms of trend and variability. See our response above and new lines 193-210..**

L201 to 208 is only one long sentence and is very difficult to read.

*** We have split the sentence**

L242-243. I suggest to better characterize the effect of past climate change on GD.

*** Done**

L245-246: need to be rewritten considering that predictions should be interpreted more caution.

*** Done. See our response above.**

L248-252: Unclear. Evolutionary assumption is not the only one to explain micro-macro continuum. It is like the evolutionary hypothesis was assimilated to the only theory linking micro-to macro evolution (ie correlation among GD and species diversity). Yet there are a panel of assumptions and it is important to mention them and to see how results from empirical study can support them or not them.

*** We agree with the reviewer that multiple theories may explain the micro-macro continuum. As mentioned above, we choose to discuss three major and basal theories that make specific predictions regarding the covariation between GD and SR / PD. These theories further constitute the basis for most eco-evolutionary theories that have been invoked to explain the global latitudinal gradients. We have partly rewritten and expanded the respective paragraph where we discuss our results with regards to the suggested hypotheses (lines 249-284).**

L257. Evolutionary assumptionà micro-macro continuum (and perhaps also for l251)

*** Done**

L259: The study on fish shows that that the evolutionary speed hypothesis is not the only one to explain the correlation GD-SD.

*** We have rewritten the respective part**

L273-274. Unclear. a positive relationship between GD and climate stability has been detected since climate stability is negatively correlated to temperature trends and variability àwhat is the effect on extinction and diversification?

*** We have rewritten the respective part of the discussion (lines 286-306). See also comments above where we clarify how climate stability is measured and the expectations regarding GD.**

L445: I feel that here we miss the information of how climate stability is related to the trend and the variability of temperature (and precipitation). From the results I assume that both trends and variability are negatively correlated to stability (?) The information comes later in the discussion. Anyway, I recommend to add more information to facilitate the interpretation of the detected relations between GD and past climatic variables, as well as assumptions / expectation.

*** We have rewritten the respective part. See new Lines 470-477.**

Reviewer #2 (Remarks to the Author):

The authors answered all the points I raised earlier. Thank you very much for considering my comments. I have no comments on this version.

Sincerely

Matheus lima-Ribeiro

*** We thank the reviewer for his work as reviewer and the very positive view on our revised version.**

Reviewers' Comments:

Reviewer #1:

Remarks to the Author:

I feel that the points raised in the previous round of review have been satisfactorily addressed.

I am very happy with all the improvements added by the authors and I like the paper. I feel that the manuscript is ready for publication.

In the introduction, the paragraph describing the theory shaping genetic patterns will be very useful for the community since it clarifies the links between the different theories. The description of the tendency among variables in models in the section results really improved the paper, and the discussion now clearly links the outputs of the paper and the different theories.

I have two minor comments:

-I feel that a sentence to interpret and discuss the main factors driving genetic variation in the summary is missing: how interpret the fact that phylogeny and specific climatic variable have more influence?

-I think that it is better to talk only of the correlations between GD and SR or PD since all variables are component of diversity and it is difficult to know which one explain the other. In brief those variables are symmetric. I suggest to modify this point thorough all the manuscript.

Minor details.

L44-45: "genetic variation is a critical component of" ♦This is not a component, this is the support, the raw material of adaptive potential

L111-112: possible to add the resolution of the grid

L152: "marginally insignificant" I suggest to change for marginally significant

L181 and everywhere: I think that it is better to talk of correlation between GD and SR of PD since the variables are symmetric, only correlation is tested.

L313: "finer scale" I would like to have the resolution of the cell here

L530: PD and SR are symmetric variables more than explanatory variables. I would not include them in the same sentence.

Reviewer #1 (Remarks to the Author):

I feel that the points raised in the previous round of review have been satisfactorily addressed.

I am very happy with all the improvements added by the authors and I like the paper. I feel that the manuscript is ready for publication.

In the introduction, the paragraph describing the theory shaping genetic patterns will be very useful for the community since it clarifies the links between the different theories. The description of the tendency among variables in models in the section results really improved the paper, and the discussion now clearly links the outputs of the paper and the different theories.

*** We thank the reviewer for all their effort and feedback that significantly improved our manuscript.**

I have two minor comments:

-I feel that a sentence to interpret and discuss the main factors driving genetic variation in the summary is missing: how interpret the fact that phylogeny and specific climatic variable have more influence?

*** We have modified the abstract to include a short interpretation of our results.**

-I think that it is better to talk only of the correlations between GD and SR or PD since all variables are component of diversity and it is difficult to know which one explain the other. In brief those variables are symmetric. I suggest to modify this point thorough all the manuscript.

*** We have modified the respective points, where appropriate, throughout the manuscript.**

Minor details.

L44-45: "genetic variation is a critical component of" → This is not a component, this is the support, the raw material of adaptive potential

*** We replaced "critical component" with "raw material".**

L111-112: possible to add the resolution of the grid

*** Done.**

L152: "marginally insignificant" I suggest to change for marginally significant

*** Done.**

L181 and everywhere: I think that it is better to talk of correlation between GD and SR of PD since the variables are symmetric, only correlation is tested.

*** Done.**

L313: “finer scale” I would like to have the resolution of the cell here

*** Done.**

L530: PD and SR are symmetric variables more than explanatory variables. I would not include them in the same sentence.

*** We replaced “explanatory” with “independent”.**